# Ice-nucleating particles active below -24 °C in a Finnish boreal forest and their relationship to bioaerosols

Franziska Vogel[1,a], Michael P. Adams[2], Larissa Lacher[1], Polly B. Foster[2], Grace C.E. Porter[2,8], Barbara Bertozzi[1,b], Kristina Höhler[1], Julia Schneider[1], Tobias Schorr[1], Nsikanabasi S. Umo[1], Jens Nadolny[1], Zoé Brasseur[3], Paavo Heikkilä[4], Erik S. Thomson[5], Nicole Büttner[1], Martin I. Daily[2], Romy Fösig[1], Alexander D. Harrison[2], Jorma Keskinen[4], Ulrike Proske[2,6,c], Jonathan Duplissy[3,7], Markku Kulmala[3], Tuukka Petäjä[3], Ottmar Möhler[1], and Benjamin J. Murray[2]

[1]Institute of Meteorology and Climate Research, Karlsruhe Institute of Technology, Karlsruhe, Germany
[2]Institute of Climate and Atmospheric Science, School of Earth and Environment, University of Leeds, Leeds, United Kingdom
[3]Institute for Atmospheric and Earth System Research/Physics, Faculty of Science, University of Helsinki, Helsinki, Finland
[4]Aerosol Physics Laboratory, Physics Unit, Faculty of Engineering and Natural Sciences, Tampere University, Tampere, Finland
[5]Department of Chemistry and Molecular Biology, Atmospheric Science, University of Gothenburg, Gothenburg, Sweden
[6]Institute for Atmospheric and Environmental Sciences, Goethe University Frankfurt, Frankfurt am Main, Germany
[7]Helsinki Institute of Physics, University of Helsinki, Helsinki, Finland
[8]School of Physics and Astronomy, University of Leeds, Leeds, UK
[a]Now at: Institute of Atmospheric Sciences and Climate (ISAC), National Research Council (CNR), Bologna, Italy
[b]Now at: Laboratory of Atmospheric Chemistry, Paul Scherrer Institute, Villigen, Switzerland
[c]Now at: Hydrology and Quantitative Water Management, Wageningen University, Wageningen, Netherlands

**Correspondence:** Benjamin J. Murray (b.j.murray@leeds.ac.uk), Franziska Vogel (f.vogel@isac.cnr.it)

**Abstract.**

Cloud properties are strongly influenced by ice formation, hence we need to understand the sources of ice-nucleating particles (INPs) around the globe. Boreal forests are known as sources of bioaerosol and recent work indicates that these dominate the INP spectra above -24 °C. To quantify the INP population at temperatures below -24 °C, we deployed a portable cloud expansion chamber (PINE) in a Finnish boreal forest from March 13, 2018 to May 11, 2018. Using the 6 min time resolution PINE data, we present several lines of evidence that INPs below -24 °C in this location are also from biological sources: (i) an INP parameterization developed for a pine forest site in Colorado, where many INPs were shown to be biological, produced a good fit to our measurements; a moderate correlation of INP with aerosol concentration larger than 0.5 µm and the fluorescent bioaerosol concentration; (ii) a negative correlation with relative humidity that may relate to enhanced release of bioaerosol at low humidity from local sources such as the prolific lichen population in boreal forests. (iii) The absence of correlation with ultra-fine particles (3.5 to 50 nm) indicates that new particle formation events are not sources of INP. This study should motivate further work to establish if the commonality in bioaerosol ice nucleating properties between spring in Finland and summer in Colorado is more generally applicable to different coniferous forest locations and times, and also to determine to what extent these bioaerosols are transported to locations where they may affect clouds.

# 1   Introduction

The cloud phase (liquid, ice or mixed-phase) is a crucial property affecting the amount of incoming solar radiation that reaches the Earth's surface (Boucher et al., 2013; Matus and L'Ecuyer, 2017). Thus, clouds have an important role in the radiative budget of the Earth, and responses of clouds to a changing climate feedback are highly uncertain (Ceppi et al., 2017; Storelvmo, 2017; Murray et al., 2021). Cloud phase is strongly influenced by the presence of aerosol particles, specifically if these aerosol particles act as ice-nucleating particles (INPs) (Hoose and Möhler, 2012; Murray et al., 2012; Kanji et al., 2017).

INPs initiate heterogeneous ice nucleation in supercooled liquid cloud droplets, that could otherwise freeze homogeneously at a temperature of approximately -36 °C (Pruppacher and Klett, 2010; Herbert et al., 2015) or trigger ice formation from the vapor phase via deposition nucleation (Vali et al., 2015). Heterogeneous ice nucleation is of high relevance in the atmosphere. For example, modelling studies have shown that the reflectivity of cold boundary layer marine clouds is strongly dependent on the INP concentration, with more INP leading to a dramatically reduced cloud albedo (Vergara-Temprado et al., 2018). Clouds resulting from deep convection are also strongly sensitive to INPs. In this case, the INP spectrum is important, where INP at around -5°C trigger the Hallett-Mossop process, whereas INP active below ≈ -25 °C are key in defining how much liquid water reaches homogeneous freezing (≈ -36 °C) and therefore the microphysical properties of anvil cirrus (Takeishi and Storelvmo, 2018; Hawker et al., 2021a, b). Measurements using ground- and space-based remote sensing tools also indicate a relationship between cloud phase and aerosol (Choi et al., 2010; Kanitz et al., 2011; Villanueva et al., 2021) and there are many cases where cloud glaciation occurs at much higher temperatures than can be accounted for by homogeneous freezing or overseeding (Radenz et al., 2021).

Given the dependence of clouds on primary ice formation and the variability of INP concentrations around the globe, cloud phase in global climate models should be tied to the INP population in the atmosphere (Murray et al., 2021). However, our ability to do this is limited by our understanding of the sources, transport, interaction with clouds and sinks of the various aerosol species that have the capacity to nucleate ice (Hoose et al., 2010; Vergara-Temprado et al., 2017; Schill et al., 2020).

To date there is no robust predictor to provide information on how ice active an aerosol particle is, so we rely on measurements that quantify which and how many atmospheric aerosol particles serve as an INP. The most discussed INP types in literature are mineral dust from low latitude deserts (Chou et al., 2011; Niemand et al., 2012; Atkinson et al., 2013) and high latitude dust (Tobo et al., 2019; Sanchez-Marroquin et al., 2020; Barr et al., 2023), marine organics and sea spray aerosol (Wilson et al., 2015; DeMott et al., 2016; Irish et al., 2017; Wilbourn et al., 2020), volcanic ash (Mangan et al., 2017; Fahy et al., 2022) and some anthropogenic emissions such as combustion ashes (Umo et al., 2015; Grawe et al., 2016) or agricultural emissions (O'Sullivan et al., 2014; Steinke et al., 2016; Hiranuma et al., 2021). The general picture is that biological INP, including primary biological particles and biogenic aerosol, are thought to be important at temperatures above approx. -20 °C, whereas mineral dust is thought to be important at lower temperatures (Tobo et al., 2013; Hill et al., 2016; O'Sullivan et al., 2018; Schneider et al., 2021; Maki et al., 2023). However, our understanding of the specific sources of these biological INPs is poor, hence field measurements in locations where there is a potential for biological INP to be released are needed in order to try to understand and ultimately represent these sources in models. In this study, we focus on conducting INP measurements

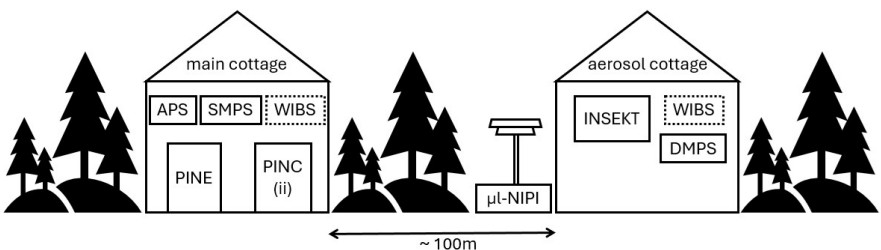

**Figure 1.** Overview of the instruments' measurement location at the SMEAR II station. Both, the main cottage and the aerosol cottage are located inside the forest. The dashed line around WIBS indicates that this instrument measured in both locations and was moved on April 3 from the main cottage to the aerosol cottage. All other instruments stayed at the same sampling location throughout the whole campaign.

in the boreal forest environment, making use of the well-established Station for Measuring Ecosystem-Atmosphere Relations (SMEAR) II located in Hyytiälä, Finland (Hari et al., 2013). Boreal forest is a known source of bioaerosols (Schumacher et al., 2013), but has not been extensively characterized in terms of its potential for producing INPs. SMEAR II is in operation since 1995, performing atmospheric and ecosystem measurements, and is well-known for frequent new particle formation events (Kulmala et al., 2001). This field site is set in a boreal forest made up of coniferous trees and is many kilometers from large urban centres or other anthropogenic aerosol sources. The HyICE-2018 campaign was launched in early 2018, a collaboration between 13 universities and institutions across 5 countries, with the aim of making ice-nucleating particle measurements in a boreal environment (Brasseur et al., 2022). The campaign was characterized by the transition period from winter to spring/-summer during which the snow, with an initial depth of up to 60 cm, melted between end of March and mid April. In the present paper we report INP measurements between -24 °C and -32 °C under mixed-phase cloud conditions (i.e. at water saturation) using the Portable Ice Nucleation Experiment (PINE) chamber (Möhler et al., 2021). This was PINE's first field deployment. The study on the same campaign presented by Schneider et al. (2021) focused on the temperature range higher than -25 °C and seasonality, while here we present measurements at temperatures below -24 °C and investigate the hourly variability. Paramonov et al. (2020) performed measurements in the same temperature range , but over a different time period. Their measurements were from end of February 2018 to beginning of April 2018, whereas ours were from mid-March until mid May, including the time of snow melt and early spring. Only four days of our measurements are included in the measurement report from Brasseur et al. (2022), challenging several literature parameterizations. In our present manuscript, we extend this analysis for the full 2 months to better understand the variability of the dataset. The focus here is on using PINE for high temporal resolution INP measurements to determine the variability of the INP concentration below -24 °C and which aerosol types contribute to the INP spectra at these low temperatures in this boreal forest environment.

## 2 Methods

 ### 2.1 The PINE chamber

PINE is a mobile cloud expansion chamber, which was developed in 2017 and commercialized in 2019 (Möhler et al., 2021). PINE has been used for both field measurements of INPs (the first example of which is presented here) and in laboratory based studies (Ponsonby et al., 2024). In this study, the first prototype version, PINE-1A, was deployed, therefore the following description is specified for PINE-1A and its configuration during the HyICE-2018 campaign.

PINE-1A consists of five interconnected parts: the inlet system, the cloud expansion chamber, the cooling system, the particle detection system and the control system. The inlet system contains a sampling tube connecting the ambient air inlet with two parallel mounted nafion drying columns (Perma Pure, MD-700-24S-1). They are a crucial part of the inlet system, because they remove humidity from the sampled air in order to prevent the inlet to the chamber to become blocked with ice. Behind the dryers, a dew point sensor (dew point mirror MBW 973) monitors the humidity of the dried sampling air. For background measurements sampling particle free air, an aerosol filter is manually installed between the ambient air inlet and the dryers. The core of PINE is the cloud expansion chamber, which has a cylindrical shape with conical end caps and a volume of approximately 7 L. Inside the chamber are three thermocouples, mounted with a 5 $\mathrm{cm}$ distance to the wall, measuring the gas temperature at the top, in the middle and at the bottom of the chamber. On the same levels are another set of temperature sensors glued to the wall to measure the temperature there. To cool the cloud expansion chamber to mixed-phase cloud temperatures, a chiller (Lauda RP 855; Lauda-Königshofen, Germany) is used, where cooled ethanol flows through pipes wrapped around the cloud expansion chamber. It can either hold the chamber at a constant temperature or perform a predefined temperature ramp given the temperature to attain and the time to hold this temperature. Mounted on the bottom of the cloud expansion chamber is an optical particle counter (OPC), which is, in this study, a combination of a welas® 2500 sensor and a Promo® 2000 control unit (Palas GmbH). The OPC detects larger aerosol particles, liquid cloud droplets and ice crystals as single particles, based on the size and shape dependent scattering signal. Due to their spherical shape, liquid cloud droplets are detected with their actual size, while the aspheric ice crystals show a higher scattering signal and are thus detected a larger diameter than cloud droplets. Aerosol particles have been proved to be detected at smaller diameters than cloud droplets and can thus be differentiated from ice crystals. A detailed discussion can be found in Möhler et al. (2021). During the HyICE-2018, the OPC was set to measure particles with a diameter between 0.6 µm and 40 µm. The welas® 2500 sensor only analyzes 10 % of the particles crossing the sensor, which leads to a detection limit of approximately 5 $\mathrm{L}^{-1}$. The control system, made of multiple flow controllers and a LabView software, ensures a smooth operation of PINE bringing all the components together.

PINE operates in a continuous way, where sequences composed of the three modes named "flush mode", "expansion mode" and "refill mode" are repeated constantly. During the flush mode the chamber is filled with air containing the aerosol under investigation, and the pressure and the gas and wall temperature are held constant. When the aerosol population inside the chamber is renewed, the expansion mode is started by closing the PINE inlet, while continuing to pump out air with a designated flow rate. By pumping out air, the pressure inside the chamber decreases adiabatically and by that also the gas temperature. Consequently, the saturation with respect to water and ice increase allowing liquid cloud droplets and ice crystals

to form once they exceed supersaturation. The expansion mode ends when the pressure reaches a pre-set value. The expansion mode is followed by the refill mode, where the chamber is slowly filled up with particle free air to the initial pressure. At the end of the refill mode, the chamber is reopened and the next flush mode begins. For each individual expansion mode an INP concentration is calculated by combining the number of ice crystal counts and the volume of analyzed air, and corrected for standard conditions (i.e. all ice-particle concentrations are reported at standard temperature and pressure) . The INP concentration is assigned to the lowest temperature measured during the expansion mode with the gas temperature sensor at the bottom of the chamber. The uncertainty for the INP concentration is given as 20 % (Möhler et al., 2021) and is not displayed in the following figures to keep them clear.

## 2.2 Installation and operation of PINE at SMEAR II

At the SMEAR II station, PINE was placed in the main cottage (Fig. 1) situated in the forest (Brasseur et al., 2022). The other online INP counters, Portable Ice Nucleation Chamber (PINC, operated by ETH Zurich; Kanji et al. (2013)) and Portable Ice Nucleation Chamber ii (PINCii, operated by University of Helsinki and University of Gothenburg; Castarède et al. (2023)), were also installed in the main cottage. Results from these complementary instruments are presented in Paramonov et al. (2020) and Brasseur et al. (2022). All instruments sampled from a heated total aerosol inlet positioned 6 m above ground level. Due to the setup of the PINE chamber in the cottage, the inlet line contained some bends that resulted in the loss of larger aerosol particles. An additional impactor was not installed. The PINE inlet was characterized onsite using an OPC (MetOne, GT 526S) directly at the aerosol inlet and at the entrance of PINE to compare the number concentration in five different size bins of particles behind the inlet with those entering PINE. With that the transmission efficiency was calculated, which showed a 50 % cut-off for particles between 5 μm and 10 μm in diameter. In the size range between 3 μm and 5 μm, 80 % of the particles were still able to enter the PINE chamber.

PINE was continuously operated from March 13, 2018 until May 11, 2018 in the temperature range of mixed-phase cloud conditions between -24 °C and -32 °C. Most days PINE ran at a constant temperature between -28 °C and -30 °C and measurements at the higher and lower ends of the broader temperature range were mainly reached while performing temperature ramps. Temperature ramps were programmed to last approximately 3 h, in which the temperature was decreased every hour by 2 to 3 °C. The cooling times to attain the temperatures were in the range of a few minutes.

In this campaign we operated PINE under mixed-phase cloud conditions, so we always generated a liquid cloud and counted the number of ice crystals that grew out of the liquid cloud. During an expansion, the temperature inside the cloud chamber is lowered around 6 °C within 40 s. It is evident that water saturated conditions are reached with the formation of a liquid cloud. Any ice crystals that form during the expansion grow rapidly in the strongly ice-supersaturated environment. In contrast to other instruments, the relative humidity during the expansion was not directly measured since the appearance of droplets defines the RH in the chamber.

Each morning a background test was performed, to ensure that no frost artefacts are seen in the OPC signal, in order to avoid miscounting ice crystals. For background tests, an aerosol particle filter is mounted between the ambient air inlet and

the dryers and the cloud expansion chamber is filled with particle free air. In this configuration, five expansions are conducted, after which no ice crystals and almost no cloud droplets are counted by the OPC, showing that the chamber is fully clean.

Throughout the campaign, PINE was operated with the same settings for the flush, expansion and refill mode which were as follows:

- Flush mode: flow of 3 L min$^{-1}$ for a duration of 4 min

- Expansion mode: flow of 4 L min$^{-1}$ until the pressure inside the chamber reaches 700 mbar

- Refill mode: flow of 3 L min$^{-1}$ until the chamber is filled

Given these settings, the total duration of one sequence of flush mode, expansion mode and refill mode was approximately 6 min.

## 2.3  Offline INP measurements

The online INP measurements of PINE are compared to two different offline INP methods, namely INSEKT (Ice Nucleation Spectrometer of the Karlsruhe Institute of Technology) and $\mu$l-NIPI ($\mu$l Nucleation by Immersed Particles Instrument). Those freezing assay techniques are described in detail in Schneider et al. (2021) and O'Sullivan et al. (2018), and only a short explanation is given here. The specific methodology employed in HyICE-2018 is also described in Brasseur et al. (2022) and the associated data are publically available in databases (https://doi.org/10.5281/zenodo.10469663 and https://doi.org/10.5445/IR/1000120666). For both methods, aerosol particles are collected on 0.4 μm or 0.2 μm nuclepore filters using a defined flow for a given time. The INSEKT filters used for comparison were sampled for 24 h in the aerosol cottage, about 100 m distant from the main cottage, and $\mu$l-NIPI filters sampled for between 4 and 18 h using commercial samplers with PM10 inlet heads (BGI PQ100, Mesa Laboratories Inc.). For the offline INP analysis, the aerosol loaded filters are washed into 8 ml of nanopure$^{\text{TM}}$ (conductivity of approximately 0.056 - 0.057 μS cm$^{-1}$) water ($V_{\text{sol}}$) to create a suspension containing the sampled aerosol. For INSEKT filter analysis, the original suspension is diluted 15 and 225 or 10 and 100 times, which allows the resulting INP-temperature spectrum to be extended towards temperatures as low as -25 °C. The suspension is placed in 50 μL volumes in the wells ($V_{\text{well}}$) of a PCR plate. As a freezing reference, a subset of wells are filled with nanopure$^{\text{TM}}$ water. The PCR plates are placed in an aluminum block, which is cooled at a rate of 0.33 °C min$^{-1}$ to a temperature at which all solution droplets inside the wells freeze. A camera records the freezing of the droplets and together with the temperature at which they froze an INP-temperature spectrum is obtained. The resulting INP concentration is calculated per standard liter of air following equation 1:

$$c_{\text{n,INP}} \;=\; \frac{V_{\text{sol}}}{V_{\text{air}}} \; c_{\text{n,INP,sol}} \;=\; -\frac{V_{\text{sol}}}{V_{\text{air}}} \; \frac{d}{V_{\text{well}}} \; \ln(f_{\text{liq}}(T)) \quad . \tag{1}$$

$d$ is the dilution factor, $V_{\text{air}}$ is the volume of the sampled air, that passed the filter and $f_{\text{liq}}$ is the fraction of liquid wells at a certain temperature of the measurement.

For $\mu$l-NIPI filter analysis, the suspension is pippetted to form an array of droplets of 1 μL volume on a hydrophobic glass slide. The glass slide is cooled with a rate of 1 °C min$^{-1}$, until all droplets freeze. As with INSEKT, a camera records the freezing of the droplets. The obtained INP concentrations are given per standard liter of air.

## 2.4 Additional aerosol and meteorological measurements

SMEAR II is an aerosol and trace gas measurement site equipped with a large set of instrumentation measuring meteorological variables and aerosol properties (Hari et al., 2013; Junninen et al., 2009; Neefjes et al., 2022). Extended descriptions of the instruments can be found in Schneider et al. (2021) and Brasseur et al. (2022), so here we briefly mention the instruments and measured variables utilized in this study.

Meteorological variables are measured at different heights above ground level on a 150 m meteorological mast. If possible,
data closest to the inlet height of 6 m are utilized, however, some variables are only available at somewhat higher or lower levels. Table 1 summarizes the instrumentation and height at which the measurements are taken. Aerosol particle size distributions are measured with a DMPS, for diameters between 3 nm and 1000 nm, and with an APS the particles between 0.5 μm and 20 μm in diameter. The data are merged during analysis. For consistency, we work with the DMPS and APS data from Schneider et al. (2021) and refer for further information to their work. A WIBS (Wideband integrated bioaerosol sensor) measures particle
fluorescence with two excitation lasers (wavelength of 280 nm and 370 nm), and emission is monitored in two detection bands (310 nm to 400 nm and 420 nm to 650 nm). A fluorescence threshold is applied to differentiate highly fluorescent biological particles from weakly fluorescent particles like dust. The two lasers and the two detection bands allow detailed differentiation of particle types (Cornwell et al., 2023). WIBS detected particles with an optical diameter larger than 0.5 μm. During the campaign a WIBS-NEO (Droplet Measurement Technology) was operated. From March 11 until April 3, it was installed in
the main cottage and thereafter moved to the aerosol cottage. The cottages have inlets with a different size cut-off, specifically 5 μm in the main cottage and 10 μm in the aerosol cottage. The concentrations of the large particles measured with WIBS show no significant increase after the change of the location, so we use the data without applying a correction. More detailed information can be found in Schneider et al. (2021) and Brasseur et al. (2022).

## 3 Results

For two months, PINE measured the INP concentration continuously in a boreal forest for temperatures between -24 °C and -32 °C, the lower temperature regime of mixed-phase cloud conditions.

Figure 2 provides an overview on how the 1 h time averaged PINE data from this monitoring (black stars) compares to the filter based measurements and CFDC measurements of PINC from the same campaign (Figure 2a) and measurements from various locations and source regions around the world (Figure 2b) acquired from filter samples, precipitation samples and
195 continuous flow diffusion chamber measurements. For the given temperatures, the PINE INP measurements span two orders of magnitude from a minimum INP concentration of approximately 0.5 L$^{-1}$ to 500 L$^{-1}$. The PINE data mainly extend the INP spectra of INSEKT and $\mu$l-NIPI towards lower temperatures. An overlap with INSEKT is only found at temperatures near -25

**Table 1.** Overview of variables measured at SMEAR II and used here for data analysis. The information are divided into the measured variable, the instrument and the sampling height above ground level.

| Variable | Instrument | Height above ground (m) |
|---|---|---|
| Wind speed | Thies 2D Ultrasonic anemometer | 8.4 |
| Wind direction | Thies 2D Ultrasonic anemometer | 8.4 |
| Temperature | Pt100 inside custom shield | 4.2 |
| Relative humidity | Rotronic MP102H RH sensor | 16.8 |
| Pressure | Druck DPI 260 barometer | ground |
| Precipitation | Vaisala FD12P weather sensor | 18 |
| Snowfall | Vaisala FD12P weather sensor | 18 |
| Snow depth | Jenoptik SHM30 | ground |
| Aerosol size distr. 3 nm to 1000 nm | Differential mobility particle sizer (DMPS) | 8 |
| Aerosol size distr. 0.5 μm to 20 μm | Aerodynamic particle sizer (APS, TSI model 3320) | 6 |
| Fluorescent particles 0.5 μm to 30 μm | Wideband integrated bioaerosol sensor (WIBS) | 6 (March 11 - April 3), 8 (April 3 - May 11) |

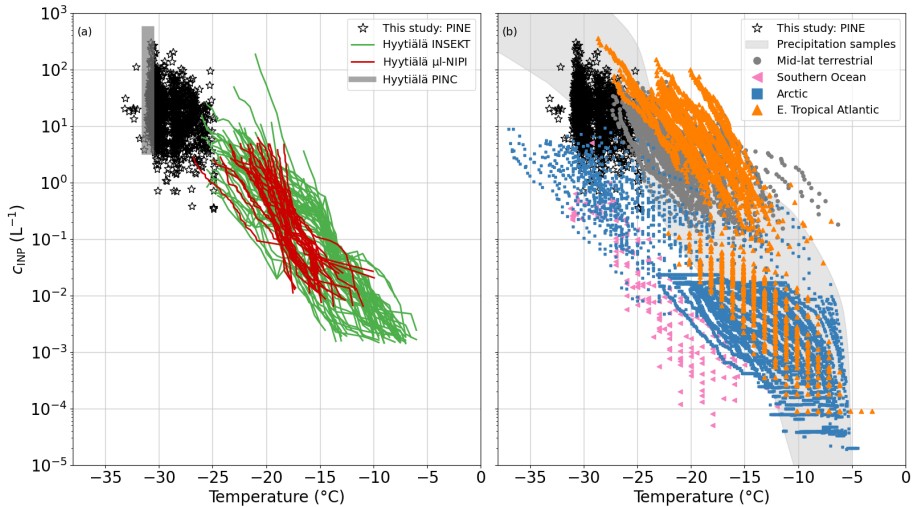

**Figure 2.** (a) Comparison of INP concentrations ($c_{INP}$) measured with PINE (black stars), INSEKT (green lines), $\mu$l-NIPI (red lines) and PINC (grey box) during HyICE-2018 and (b) with INP measurements from a variety of locations around the world. The precipitation samples collected around the world (grey shading) originate from Petters and Wright (2015), the Arctic measurements (blue squares) are from Irish et al. (2019); Wex et al. (2019); Porter et al. (2022), those from the eastern tropical Atlantic (orange triangles) are from Price et al. (2018); Welti et al. (2018), the Mid-latitude terrestrial data (grey dots) is from O'Sullivan et al. (2018) and data from the southern ocean (pink triangles) from McCluskey et al. (2018).

°C. PINE and PINC data overlap at the lower end of the covered temperature range and fully agree in the span of measured INP concentrations. Therefore, once combined, data from HyICE-2018 provide INP data from the Finnish boreal forest between

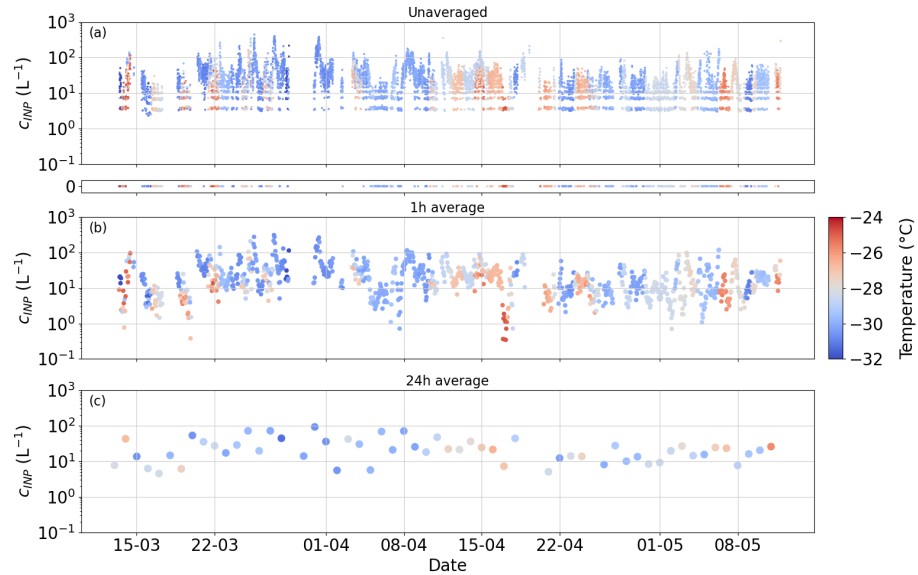

**Figure 3.** Time averaged PINE INP data over different periods. The measurements cover freezing temperatures between -24 °C and -32 °C. (a) Shows the unaveraged PINE data, where each data point represents one expansion. The small panel below indicates the expansions in which no ice crystals were counted. (b) and (c) contain the 1 h and 24 h time averages, respectively.

-6 °C and -32 °C and thus cover almost the full mixed-phase cloud regime. In comparison to other locations, the PINE INP data compare best with INPs from mid-latitude terrestrial sources (grey dots). INP concentrations measured in the Arctic (blue squares) and southern ocean (pink triangles) are mainly lower, whereas INP measurements from the eastern tropical Atlantic (orange triangles) are mainly higher than what was measured in the boreal forest. So the INP concentration measured with PINE in Hyytiälä falls within the middle range of INP concentrations in comparison to a more global overview.

### 3.1 Time averaging PINE data

PINE measured INP concentrations with a time resolution of approximately 6 min, where from each expansion one data point was obtained (Figure 3a). The range of the unaveraged measured INP concentrations spreads over two orders of magnitude between 5 L$^{-1}$ and approximately 500 L$^{-1}$. These single expansion data show the same variation over the entire two months of the campaign, where very high and very low INP concentrations were measured within a few hours or days. Among all the performed expansions, about 10 % had 0 ice crystal counts and are therefore represented as an INP concentration of 0 L$^{-1}$. These 0 ice crystal runs occurred throughout the campaign and over almost the entire temperature range between -24 °C and -32 °C emphasizing the observed variation in the overall INP concentration. A quantisation of the data is visible, when only 1, 2 or 3 ice crystals were counted during an expansion as stripes of data points at around 5, 8 and 10 L$^{-1}$, respectively, carrying a large counting error of $\sqrt{n}$. However, the daily background tests ensured that these low ice counts are associated with INP, rather than frost artefacts from the chamber walls or other artefacts such as electrical noise. A detailed discussion of

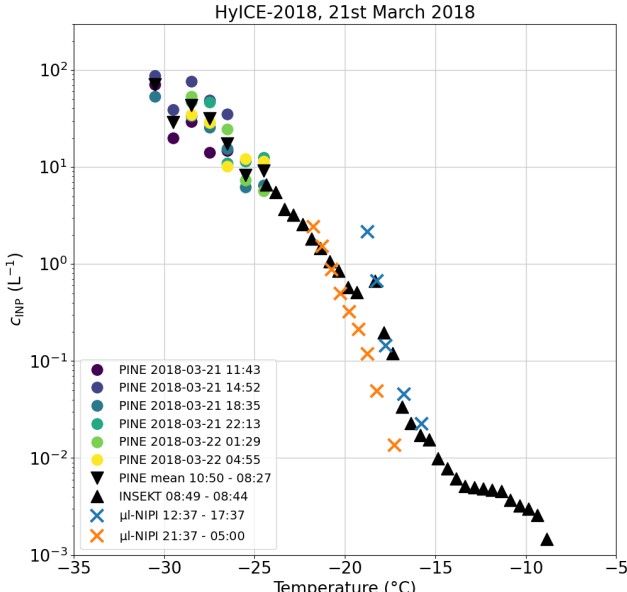

**Figure 4.** Comparison of the INP concentration of PINE temperature ramps with INSEKT and $\mu$l-NIPI data in the sampling period of March 21. For the sampling period of the INSEKT filter (black triangles) the start and stop time are given, with the start day on March 21 and the stop day on March 22. The $\mu$l-NIPI filters are divided into a day (orange cross) and a night (blue cross) filter. PINE measurements are split to the individual temperature ramps (colored dots) and an average of the full sampling period of PINE (black upside down triangles). The times of the individual PINE temperature ramps are the times when half the ramp is over.

the background measurements is given in Möhler et al. (2021). The non-averaged high time resolution data illustrate short-term INP variability, but to improve counting statistics and compare with other data, some time averaging is needed. To improve the counting statistics, we average over several expansions, thus also reducing the influence of instances where there are zero, or very few ice crystals observed per expansion. With 1 h averages all zeros and quantized values are removed, while information on the temporal evolution of the INP concentration is preserved (Figure 3b). In addition, 1 h averages lead to an order of magnitude increase in the lowest detectable INP concentration. To compare the PINE measurements with offline INP methods such as INSEKT and $\mu$l-NIPI, even longer time averaging, in this case 24 h, is needed (Figure 3c). By averaging such highly time resolved data over one day, information about the short-term variation on a time scale of hours gets lost and only trends on a longer time scale of days to weeks are pictured. This highlights that INP measurements with a high time resolution, as done with PINE, over a long time period are important. Various time averages can be applied later, depending on the goal of the analysis.

### 3.2 Comparison of temperature ramps of PINE with INSEKT and $\mu$l-NIPI

The consequences on the time averaging and the loss of short-term variations in the INP concentration can also be seen by comparing PINE measurements to INSEKT and $\mu$l-NIPI (Figure 4). An INSEKT filter was typically sampled for 24 h, while

µl-NIPI filters were sampled for 12 h (day/night) and PINE performed in the same time period six temperature ramps, where one ramp lasts approximately 3.5 h. In this case three temperatures were set during the ramp resulting in data points scattered by ± 2 °C around the set temperature. To obtain a continuous spectrum, the data were binned in 1 °C steps. The overlap in temperature for INSEKT and PINE is only at -24 °C while PINE and µl-NIPI do not overlap. However, PINE nicely extends the INP-temperature spectrum towards lower temperatures. µl-NIPI measurements overlap with INSEKT for the entire temperature

range. Comparing the day and night filters from µl-NIPI, one can see that the day filter is fully in the range of the INSEKT filter for temperatures higher than -18 °C, and the night filter shows up to one order of magnitude lower INP concentrations for temperatures between -18 °C and -20 °C. During the same sampling period, PINE performed six temperature ramps, which show a variation of a factor of 5 over the captured temperature range. The PINE data at -24 °C is consistent with the INSEKT data at that temperature. On average, the PINE measurements (black upside down triangles) align well with the INSEKT INP-

temperature spectrum and extend these measurements towards lower temperatures. This comparison is another illustration that PINE is able to capture a short-term variability which would otherwise be missed with the filter-based measurements or when averaging data.

### 3.3 Comparison of measured INP concentrations to those predicted with literature parameterizations

To represent primary ice formation in the atmosphere in models, a number of parameterizations have been developed and

245 relate the INP concentration to various properties of the aerosol population or to meteorological variables. Here, we compare our PINE INP measurements with four parameterizations that we consider the most suitable for the measurement location. Those are from DeMott et al. (2010), based on measurements in various locations and different aerosol types, Tobo et al. (2013), who used data acquired in a conifer forest in summertime to present two different parameterizations, and Schneider et al. (2021) making use of measurements during the same campaign using INSEKT data. Hereafter, the four parameterizations

are referred to as DeMott 2010, Tobo 2013 (1), Tobo 2013 (2) and Schneider 2021. DeMott 2010 and Tobo 2013 (1) base their INP concentration predictions on the concentration of aerosol particles larger than 0.5 µm in diameter, Tobo 2013 (2) uses the concentration of fluorescent particles with a diameter larger than 0.5 µm measured with a UV-APS with an excitation wavelength of 355 nm and an emission wavelength range between 420 nm and 575 nm and Schneider 2021 found the ambient air temperature to be a predictive parameter. The valid temperature ranges for DeMott 2010 and Tobo 2013 (1) and (2) are

given from -15 °C to -35 °C and -5 °C to -35 °C, respectively, covering the full temperature range of the PINE measurements. Even though fluorescent particle measurements to develop the Tobo 2013 (2) parameterization were done with a different instrument, it is interesting to apply this parameterization on our data due to the similarity of the measurement environments. Due to the setup of WIBS it provides measurements of seven different combinations of excitation wavelength and detection band wavelength, whereas the UV-APS provides only one channel. By that the WIBS can capture more various bioaerosols

and the INP concentrations predicted by Tobo 2013 (2) should be seen as an upper estimation. Schneider 2021 is limited to temperatures between -5 °C and -25 °C, which is mainly outside the range of the PINE measurements. However, since it is based on measurements in the same location and during the same time period, it is still valuable to apply this parameterization to our data.

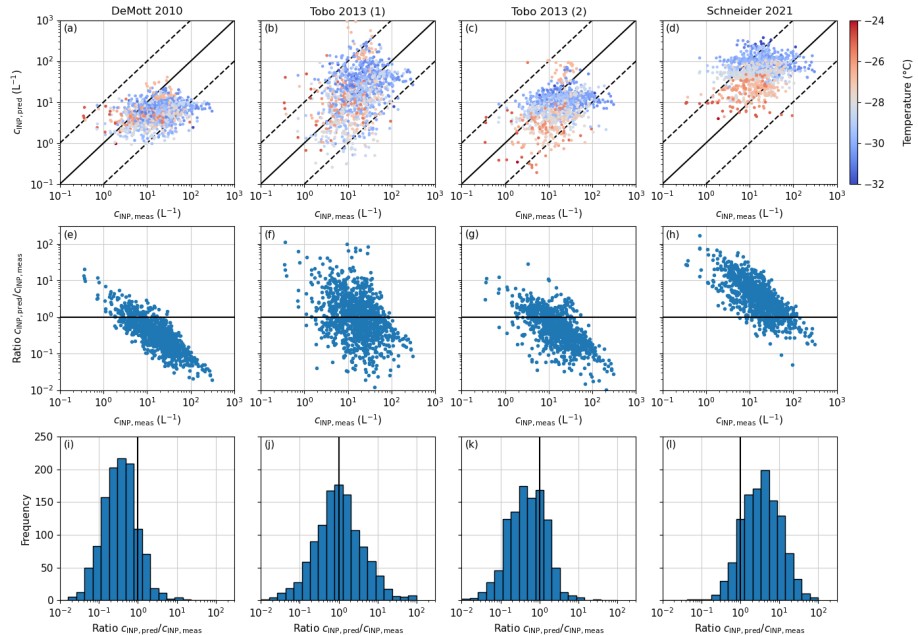

**Figure 5.** Predictions for PINE INP concentration using the parameterizations of DeMott 2010 (a), Tobo 2013 (1) and (2) (b) and (c) and Schneider 2021 (d). The solid line represents the 1:1 line and the dashed lines are a factor of 10 lower and higher than the 1:1 line. The plots in the second row ((e), (f), (g) and (h)) represent the evolution of the ratio of the predicted INP concentration ($c_{\mathrm{INP,pred}}$) to the measured INP concentration ($c_{\mathrm{INP,meas}}$) with an increasing measured INP concentration. The solid line is unity, where the prediction matches exactly the measured value. (i), (j), (k) and (l) are histogram representations of the ratio $c_{\mathrm{INP,pred}}$ / $c_{\mathrm{INP,meas}}$, where the solid line is again unity.

Applying DeMott 2010, Tobo 2013 (1) and (2) and Schneider 2021 to our PINE data to test them for their predictive capability, 1 h time averaged data were used to align the INP data with the aerosol data from the SMEAR II APS. DeMott 2010 tends to underpredict the majority of the data by a factor of 5 to 10 (Figure 5a). In order to highlight the offset from the 1:1 line, the ratio of the predicted to the measured INP concentration ($c_{\mathrm{INP,pred}}$ / $c_{\mathrm{INP,meas}}$) was calculated, where one is a perfect prediction, values larger than one indicate an overprediction of the parameterization and smaller values an underprediction. In the case of DeMott 2010, a clear trend from overprediction to underprediction is visible for increasing INP concentrations (Figure 5e). This corresponds to the temperature dependence of the parameterization being too shallow. The corresponding frequency histogram in Figure 5i peaks between 0.5 and 0.7 showing that the majority of the data points remains within a factor of 10. Moreover, DeMott 2010 gives a variation in the INP concentration of generally less than two orders of magnitude, whereas the measured INP concentrations vary by almost three orders of magnitude.

Comparing our PINE INP data with Tobo 2013 (1) predictions, the data are scattered around the 1:1 line with the majority of data within a factor of 10 (Figure 5b). For Tobo 2013 (1), the $c_{\mathrm{INP,pred}}$ to $c_{\mathrm{INP,meas}}$ ratio indicates a slight trend towards an underprediction for higher INP concentrations (Figure 5f), but less pronounced than for DeMott 2010. The frequency histogram in Figure 5j shows a symmetric distribution around unity, underlining that overall Tobo 2013 (1) predicts the INP concentration

well, which could be because it is based on measurements in a coniferous forest with similar biological INP sources to those in the boreal forest in Finland.

When using the concentration of fluorescent particles as a predictive parameter in Tobo 2013 (2), the INP concentration tends to be underpredicted by about a factor of 2 (Figure 5c). As discussed before, this is the upper estimate of the INP concentration due to the differences in UV-APS and WIBS, and the predicted INP concentrations can be even lower. As for DeMott 2010, low INP concentrations were more overpredicted than higher INP concentrations (Figure 5g). The histogram (Figure 5k) has a wide peak between 0.5 and 1. This means that the measured INP concentration from the boreal forest in Finland cannot be predicted by biogenic particles (as measured by the WIBS) only, but is rather a combination of biogenic particles and larger aerosol particles from different sources.

In contrast to DeMott 2010 and Tobo 2013 (2), Schneider 2021 overpredicts the INP measurements, with larger deviations at lower temperatures (about one order of magnitude below -28 °C; Figure 5d). At temperatures that overlap with the INSEKT data (-24 to -25 °C) the agreement between the parameterization and PINE measurements is much better. Compared to DeMott 2010 and Tobo 2013, Schneider 2021 links INP concentration to the ambient air temperature to represent the seasonality in the INP concentration. The ratio of predicted to measured INP concentration in Figure 5g shows a clear overprediction for lower INP concentrations and an underprediction for the higher measured INP concentrations. Also the frequency plot in Figure 5l shows a symmetric distribution with a peak between five and seven. The discrepancies applying the Schneider 2021 parameterization may originate from the valid temperature range, which is above -25 °C. Furthermore, the INP data measured with PINE have a less pronounced temperature dependence compared to those measured with INSEKT, which could be due to a different portion of the aerosol particle population serving as INP in the respective temperature regimes. Moreover, freezing assays like INSEKT measure only INPs from the immersion freezing mode, while PINE can also capture deposition and condensation nucleation. At temperatures higher than -36°C, the temperature at which water freezes homogeneously, the contribution of deposition/condensation nucleation is thought to be low (Westbrook and Illingworth, 2011), but may still explain some of the discrepancies.

## 3.4 Correlation of INP concentration with aerosol and meteorological variables

Driving factors for changes in the INP concentration can vary, as is hinted at in the application of the different predictive parameters used for the DeMott 2010, Tobo 2013 (1) and (2) and Schneider 2021 parameterizations. Only the Tobo 2013 (1) parameterization predicted the measured INP concentration within one order of magnitude. To further investigate potential connections between ice nucleation activity and meteorological variables and aerosol properties at temperatures between -27 °C and -30 °C, correlation coefficients are calculated. The correlations are calculated using the Spearman correlation coefficient. Here, we also present the p-value connected to each correlation coefficient, indicating whether a correlation is statistically significant ($p < 0.05$) or not ($p > 0.05$). Correlations for all meteorological variables listed in Table 1, as well as some aerosol properties and 'time over land' and the INP concentration are displayed in Figure 6. The time over land is a parameter determined from back-trajectory analysis and corresponds to the time an air mass spends over the boreal forest environment before being measured at the SMEAR II station. For the correlation analysis, only air masses from the corridor between

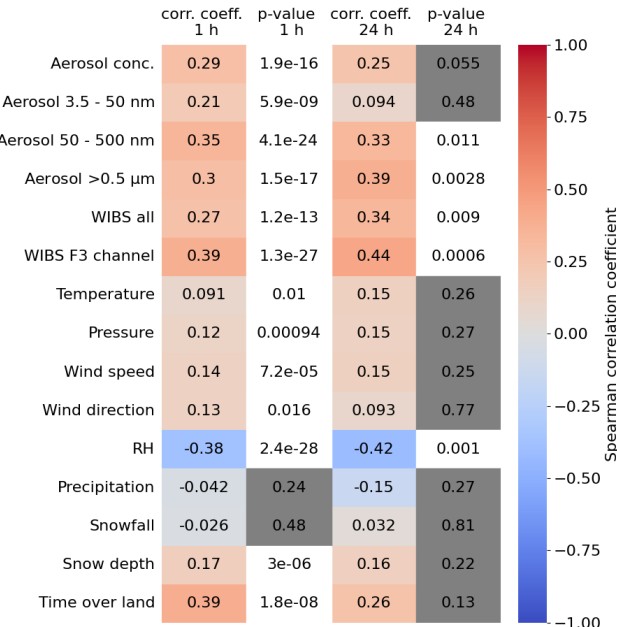

| | corr. coeff. 1 h | p-value 1 h | corr. coeff. 24 h | p-value 24 h |
|---|---|---|---|---|
| Aerosol conc. | 0.29 | 1.9e-16 | 0.25 | 0.055 |
| Aerosol 3.5 - 50 nm | 0.21 | 5.9e-09 | 0.094 | 0.48 |
| Aerosol 50 - 500 nm | 0.35 | 4.1e-24 | 0.33 | 0.011 |
| Aerosol >0.5 µm | 0.3 | 1.5e-17 | 0.39 | 0.0028 |
| WIBS all | 0.27 | 1.2e-13 | 0.34 | 0.009 |
| WIBS F3 channel | 0.39 | 1.3e-27 | 0.44 | 0.0006 |
| Temperature | 0.091 | 0.01 | 0.15 | 0.26 |
| Pressure | 0.12 | 0.00094 | 0.15 | 0.27 |
| Wind speed | 0.14 | 7.2e-05 | 0.15 | 0.25 |
| Wind direction | 0.13 | 0.016 | 0.093 | 0.77 |
| RH | -0.38 | 2.4e-28 | -0.42 | 0.001 |
| Precipitation | -0.042 | 0.24 | -0.15 | 0.27 |
| Snowfall | -0.026 | 0.48 | 0.032 | 0.81 |
| Snow depth | 0.17 | 3e-06 | 0.16 | 0.22 |
| Time over land | 0.39 | 1.8e-08 | 0.26 | 0.13 |

**Figure 6.** Spearman correlation coefficient calculated for full campaign period using 1 h and 24 h time averages of the PINE INP data. The temperature range of PINE measurements was limited to temperatures between -27 °C and -30 °C. Red colors represent a positive correlation, whereas blue colors indicate a negative correlation. The p-values marked with grey boxes indicate p > 0.05, i.e. the correlation is not considered to be statistically significant.

north and west of Hyytiälä, the so-called clean sector, are considered to eliminate potential continental aerosol sources. More information concerning this parameter and the way it is calculated can be found in Petäjä et al. (2022). To limit biases in the correlations due to the wide temperature range of the PINE measurements, only data points between -27 °C and -30 °C are
315 considered, which includes the majority of all data and covers the entire time range. In Figure 6, we show the correlation coefficients for 1 h averaged data as well as the 24 h averaged data. The correlation coefficients are generally not significantly different between the two time averages, however, the p-value gives more statistical significance to the 1 h averages, so we focus the discussion on that data.

The highest positive correlations, albeit still weak, are found between INP concentration and the WIBS F3 channel (0.39)
and the time over land (0.39). The p values indicate that these correlations are significant. An aerosol particle entering WIBS gets excited by two lasers with a different wavelength and can be detected in two different detection bands. Depending on its composition, fluorescence is detected in only one or more laser detection band channels. The F3 channel is fluorescence triggered by the 370 nm laser and detected in the 420 nm to 650 nm band. Particles detected only in this channel can be assigned to NAD(P)H, which is a tracer for the viable biological fraction (Savage et al., 2017). Also the correlation between
particles larger than 0.5 µm in diameter and the total concentration of fluorescent particles show enhanced positive correlations (0.3 and 0.27). These weak correlations indicate that aerosol particles larger than 0.5 µm that are biological may preferentially

serve as INP. This is consistent with the heat test data reported by Schneider et al. (2021) for the same location where they observed a substantial decrease in the INP concentration after heating, indicating the presence of a population of protein based ice nucleating entities (Daily et al., 2022). This is also consistent with the size resolved INP measurements reported by Porter et al. (2022) who demonstrated that below -23 °C, the 2.5 to 10 μm size category was more important than smaller sized aerosol to the INP population. The correlation with aerosol concentrations between 3.5 nm and 50 nm is much weaker (0.2). There were new particle formation (NPF) events during the HyICE-2018 campaign (Brasseur et al., 2022) that created high concentrations of aerosol particles smaller than 50 nm, hence the lack of correlation with those aerosol particles indicates that they do not serve as INPs. The total aerosol concentration and the aerosol concentration of particles with diameters between 50 nm and 500 nm have values of 0.29 and 0.35, respectively, meaning that besides the bioaerosol particles from the coarse mode, there may also be INP in the population of aerosol particles smaller than 500 nm.

The correlations with meteorological variables are generally low (correlation coefficients of <0.17), with the exception of a moderate negative correlation with relative humidity (-0.38). This means that at low relative humidity, the INP concentration was generally greater and vice versa. This might be explained by the known mechanisms of bioaerosol release at low RH (Marshall, 1996; Tormo et al., 2001). Studies of the ice nucleating ability of lichens from Hyytiälä and elsewhere in the world show that several common lichen species harbour large quantities of very active ice nucleating entities (Moffett et al., 2015; Eufemio et al., 2023; Proske et al., 2024). It is also thought that bioaerosol released from lichens is enhanced at low RH (Armstrong, 1991; Tormo et al., 2001). Another study suggests the surface of Scots pine trees, the predominant tree species in Hyytiälä (Kokkila et al., 2002) as a potential source of INPs (Seifried et al., 2023). Since lichens and the bark and branches of trees are one of the few biological entities not covered in snow, they should be considered a candidate source of bioaerosol in the boreal forest. The evidence from the correlation with both aerosol properties and relative humidity is consistent with a biological contribution of INP to the INP population active at temperatures between -27 °C and -30 °C.

### 3.4.1 INP concentrations during new particle formation events

The low correlation between INPs and the concentration of small particles (Fig. 6) suggests that particles formed during new particle formation (NPF) events do not contribute to the INP population. During NPF events, gaseous species nucleate to form a critical cluster which then grows via gas to particle conversion to particles of tens of nanometers in diameter over the course of hours (Kulmala et al., 2001). These newly formed particles are very high in number concentration and can be detected based and the aerosol size distribution and the increase in aerosol number concentration. SMEAR II is known for its NPF events and the analysis and categorization of these events is based on Dal Maso et al. (2005), who introduced category Ia, Ib and II events, which designates the intensity of an event. The occurrence of NPF days during HyICE-2018 was already presented in Brasseur et al. (2022), where individual days were flagged as NPF days, independent of their category. To investigate the connection between NPF events and INPs, we flagged the hourly INP data measured with PINE at a freezing temperature between -27 °C and -30 °C with the different NPF event categories. In case none of the three NPF categories was detected, we set the flag to no NPF event. By comparing the INP concentration on NPF event days (independent of the category) and non-NPF event days

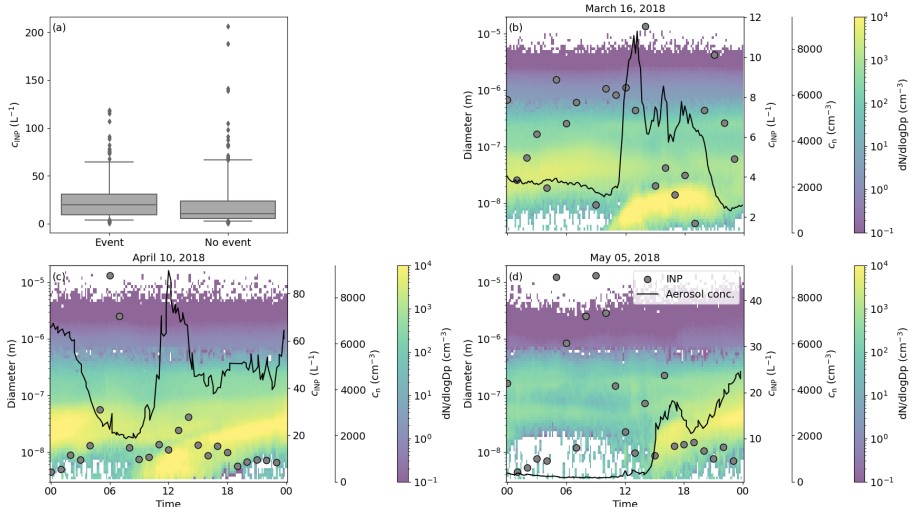

**Figure 7.** Overall comparison of the INP concentration during NPF events and outside NPF events together with three case studies. (a) Boxplot of the INP concentration on days flagged NPF event days and days without an NPF event. The borders of the box represent the 25th and 75th percentile, the horizontal line inside the box the median, the whiskers the 5th and 95th percentile and the single points all the data outside of that. (b), (c) and (d) are three cases (March 16, April 10 and May 5) of NPF events of category Ib, where the background of each panel shows the particle number size distribution, with the diameter in the left y-axis and dN/dlogDp as the colorbar. The first right y-axis is the INP concentration and the second right y-axis the aerosol number concentration measured by the DMPS and APS. Please note that the y-axis of the INP concentration varies, due to the high variability of INP.

(Fig.7a), a similar median and variability of the INP concentration is seen in the two boxes. This indicates that NPF events have no effect on the INP population.

In order to explore the NPF events in more detail, we plot time series of PINE INP measurements alongside aerosol concentrations and the particle number size distribution on three days, one from each month of the campaign. The case on March 16 (Fig.7b) shows a steep increase in the aerosol number concentration at 12:00 local time from about 2000 cm$^{-3}$ to ap-
365 proximately 8500 cm$^{-3}$, which characterizes the start of a NPF event, and is also visible in the high concentration of small particles at the same time. The aerosol concentration decreases to the initial value at 23:00 local time. During the whole day the INP concentration is scattered between 2 L$^{-1}$ and 12 L$^{-1}$ without showing a distinct pattern. In the other two cases on April 10 and May 5 (Fig.7c and d) the aerosol concentration also shows a steep increase when NPF sets in and lower aerosol concentration outside the NPF event. In these two cases the INP concentration reaches higher values during phases of lower
aerosol concentrations and drops when NPF is initiated. This may be because NPF events are favored when the aerosol surface area concentration is low, which is also when the INP concentrations would generally be expected to be lower. These findings are consistent with literature reports that found no observable nucleation of ice from secondary organic aerosol particles in the mixed-phase cloud regime (Prenni et al., 2009; Frey et al., 2018). Hence, this analysis strengthens the conclusion that particles formed during NPF events in the size range up to 50 nm in diameter do not contribute to the INP population.

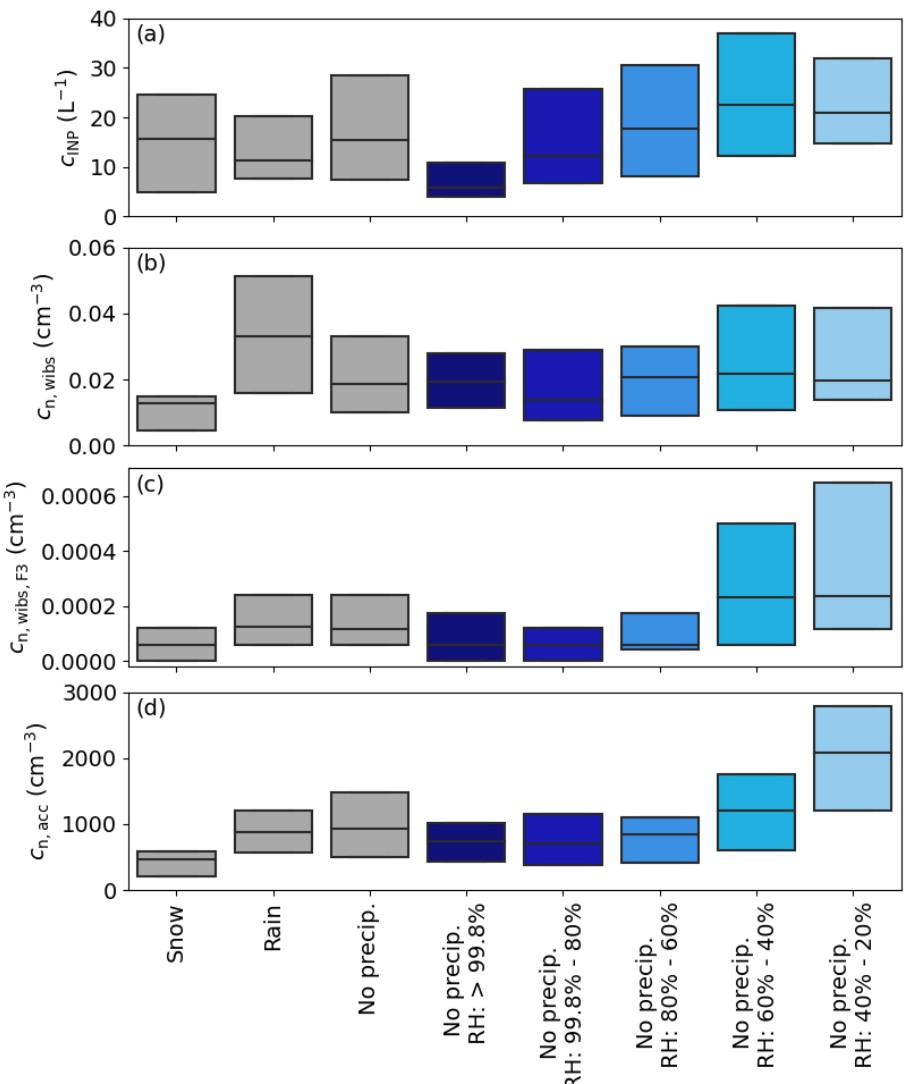

**Figure 8.** The relationship between aerosol, precipitation and humidity. (a) The INP concentration, (b) fluorescent particle concentration measured by WIBS, (c) fluorescent particle concentration measured in the F3 channel of WIBS and (d) the particle concentration in the accumulation mode are divided into several precipitation and relative humidity categories. The precipitation is split into cases of snow fall, rain fall and no precipitation (including also no snowfall). The data of the 'no precip.' category were again divided into different relative humidity (RH) classes, which are RH > 99.8 % (saturated conditions, considering some uncertainty of the measurements), 99.8 % > RH > 80 %, 80 % > RH > 60 %, 60 % > RH > 40 %, 40 % > RH > 20 %. The boundaries of the boxes are the 25th and 75th percentile. For clarity, whiskers and outliers are not shown.

### 3.4.2   The link between high INP concentrations and low relative humidity

In Section 3.4 we discuss the observation of a negative correlation between the INP concentration and the relative humidity (RH), which is potentially linked to an increased release of bioaerosol from lichen (Armstrong, 1991; Tormo et al., 2001). To better understand potential connections between the INP concentration, relative humidity and aerosol sources, the data are grouped based on precipitation into the three categories snow fall, rain fall, and no precipitation (snow fall and rain fall). The data of the last category are further split into five different RH ranges: > 99.8 % (which considers saturated conditions including the uncertainty of measurement), 99.8 % > RH > 80 %, 80 % > RH > 60 %, 60 % > RH > 40 %, 40 % > RH > 20 % (Fig.8a).

(Fig.8a) reveals no substantial change in the INP concentration between snow fall, rain fall, and no precipitation. However, when dividing the data points without precipitation into the RH categories, the lowest INP concentrations with a median of 5 $L^{-1}$ are measured for the highest RH. For the following groups the median INP concentration increases until it reaches 20 $L^{-1}$ at a RH of 60 % and lower. A similar trend with RH is observed for the aerosol concentration in the accumulation mode and the WIBS F3 channel. The WIBS F3 channel can point to increased concentration in NAD(P)H (Savage et al., 2017), a co-enzyme linked to energy metabolism in cells and is an indication for living biological organisms. In contrast, the total fluorescent particles concentration of WIBS showed no clear increase in the mean concentration with decreasing RH (Fig.8b). In other locations fluorescent bioaerosol concentrations were found to increase markedly with increasing RH, which may be related to RH-dependent fungal spore release mechanisms (Toprak and Schnaiter, 2013; Gabey et al., 2010; Timothy P. Wright and Petters, 2014). Hence, the bioaerosol released in the Boreal forest of Southern Finland appears to be different to other locations.

The WIBS results and the correlation with RH are consistent with release of bioaerosol particles from lichens. Lichens can reproduce asexually through the production of diaspores. These diaspores contain living components of lichen, the mycobiont and photobiont, that can then colonise new locations (Hale, 1974). These diaspores would therefore contain NAD(P)H since they are metabollically active. In addition, as mentioned above, release of diaspores has been shown to be negatively correlated with RH (Armstrong, 1991; Tormo et al., 2001). Hence, the hypothesis that lichen derived bioaerosol contribute to the INP population is consistent with both the F3 channel (NAD(P)H) of WIBS and the negative RH dependence.

A remarkable feature of Fig.8 is that for the fluorescent particle concentration, the WIBS F3 concentration and the aerosol concentration in the accumulation mode, the lowest values are detected during snowfall, while for the INP concentration this is not the case. This implies that many aerosol types are preferentially removed during snow fall events, but the INP concentration is not affected to the same extent. When snowing, the ambient RH would be expected to be well below 100% (with respect to water), for example, if the air were at ice saturation the RH would be about 90% at -10°C, conditions under which we might expect to start to see lichens producing aerosol. However, the low bioaerosol concentration from the WIBS is inconsistent with this idea.

During rain fall, an enhanced concentration of fluorescent particles is measured by WIBS, however it is not reflected in the INP concentration. Previous studies reported an increase of INPs during and after rain events during measurements in Colorado in summer. The highest increases in fluorescent particle and INP concentrations were observed during intense rain events with

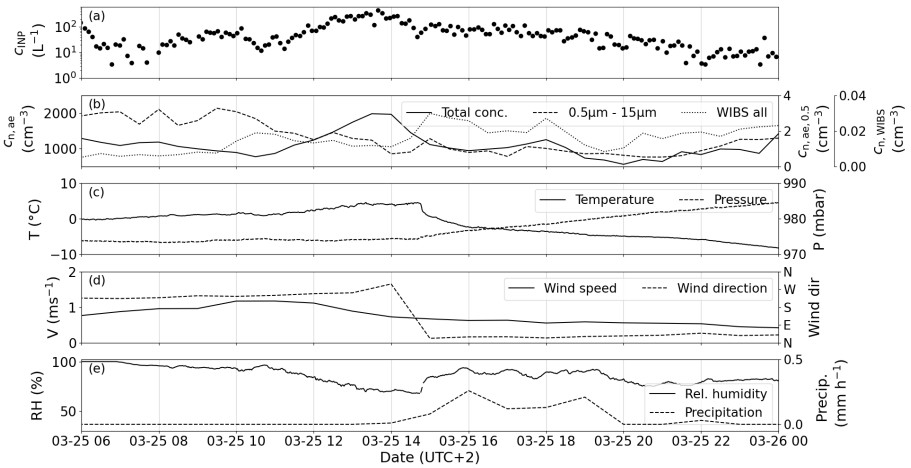

**Figure 9.** March 25 case study, where a cold front passed over Hyytiälä in the early afternoon. (a) Measured PINE INP concentration. (b) The total aerosol number concentration (solid line, left axis) and the number concentration of aerosol particles larger than 0.5 μm in diameter (dashed line, right axis) and the total fluorescent particles measured with WIBS (dotted line, secondary right axis). The ambient air temperature T (solid line, left axis) and pressure p (dashed line, right axis) are plotted in (c). (d) Wind speed v (solid line, left axis) and the wind direction (dashed line, right axis). (e) Relative humidity RH (solid line, left axis) and the accumulated snow fall (dashed line, right axis).

rain rates more than 3 mm/5min (Tobo et al., 2013). Less pronounced increases were observed during less intense events and

410 at lower measurement temperatures, which are equal to our covered temperature range (Huffman et al., 2013; Prenni et al., 2013). The overall rain rate during our measurement period was about 0.03 mm/5min and by that two orders of magnitude lower than during the measurements in Huffman et al. (2013); Prenni et al. (2013); Tobo et al. (2013). Moreover, they report measurements in summer, when rain fell on dry soil and by that released the biological particles. During the HyICE-2018 campaign, the soil was always covered by snow or wet, hence release of biological particles might have been hindered.

**3.5 Response of the INP concentration to changing ambient conditions**

In this section we present two independent case studies to examine how the measured INP concentration varied with a range of parameters. During the first case (18 h from March 25 06:00 to March 26 00:00) there was a synoptic air mass change associated with a weather front, whereas during the second case (22 h from May 02 00:00 to May 02 22:00) there was no major air mass change.

The first case (Figure 9) is characterized by a change in the overall synoptic situation induced by a cold front passing over Hyytiälä at around 14:00. Cold front passages are characterized by a drop in temperature, an increase in pressure (Figure 9c), a prompt change in wind direction (Figure 9d) and precipitation typically beginning after the change in wind direction (Figure 9e). In the hours prior to the passage of the front, the INP concentration increased by approximately two orders of magnitude from 5 L$^{-1}$ to almost 400 L$^{-1}$, falling back to lower values ($\approx$ 100 L$^{-1}$) after the front passed (Figure 9a). The

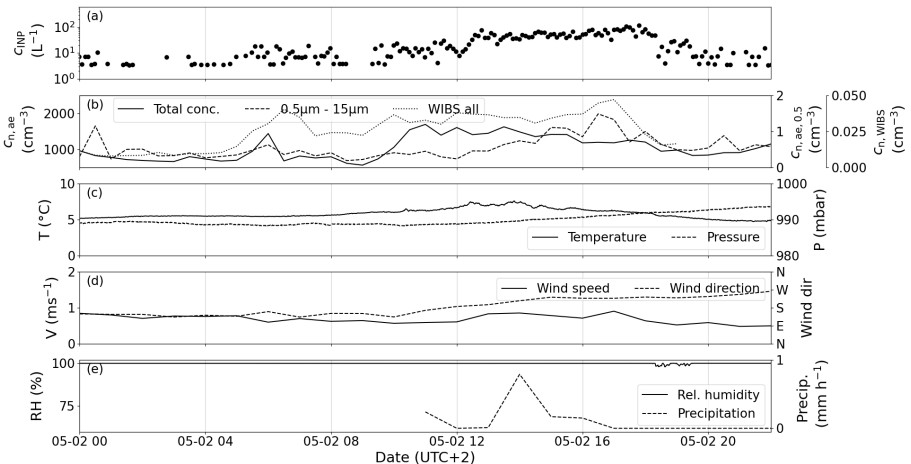

**Figure 10.** May 2 case study. (a) Measured PINE INP concentration. (b) The total aerosol number concentration (solid line, left axis) and the number concentration of aerosol particles larger than 0.5 µm in diameter (dashed line, right axis) and the total fluorescent particles measured with WIBS (dotted line, secondary right axis). The ambient air temperature T (solid line, left axis) and pressure p (dashed line, right axis) are plotted in (c). (d) Wind speed v (solid line, left axis) and the wind direction (dashed line, right axis). (e) Relative humidity RH (solid line, left axis) and the accumulated snow fall (dashed line, right axis).

INP concentration continued to decrease steadily through the afternoon and evening, perhaps due to scavenging by the snow that fell during this period. A similar behavior of the INP concentration was observed in the 30 min time resolved PINC data (Paramonov et al., 2020). The total particle concentration peaks at the same time as the INP concentration (Figure 9b), while the concentration of particles larger than 0.5 µm is more constant with higher values in the morning hours. The fact that the increase in INP concentration mirrors the peak in total aerosol concentration suggests that the additional INP are less than 0.5

430 µm. The total concentration of fluorescent particles measured with WIBS shows a small increase in concentration between 15:00 and 18:00 and with that has a profile independent of the INPs, indicating that the additional biogenic aerosol detected by the WIBS does not contribute to the INP concentration in this occasion. Inspection of Figure 9e reveals that the relative humidity decreased from approx. 90 % to approx. 70 % during the period of elevated INP concentrations. The increase in the INP concentration is perhaps related to a release of aerosol particles at decreased relative humidity, perhaps consistent with

435 the release of sub 500 nm aerosol from lichen sources, where it was observed that here was a significant negative correlation between relative humidity and INP.

In the second case, on May 2, the INP concentration varied by approximately two orders of magnitude between 5 $L^{-1}$ and 100 $L^{-1}$ (Figure 10a), without any abrupt changes in ambient air temperature, pressure or wind direction that might be associated with a front (Figure 10c and 10d). The number concentration of particles larger than 0.5 µm in diameter (Figure 10b)

increases from approximately 0.5 $cm^{-3}$ in the night and morning hours to 1.5 $cm^{-3}$ in the afternoon at 17:00. In comparison to the first case, the WIBS particle concentration increases during the day, mirroring the INP concentration. Thus changes in

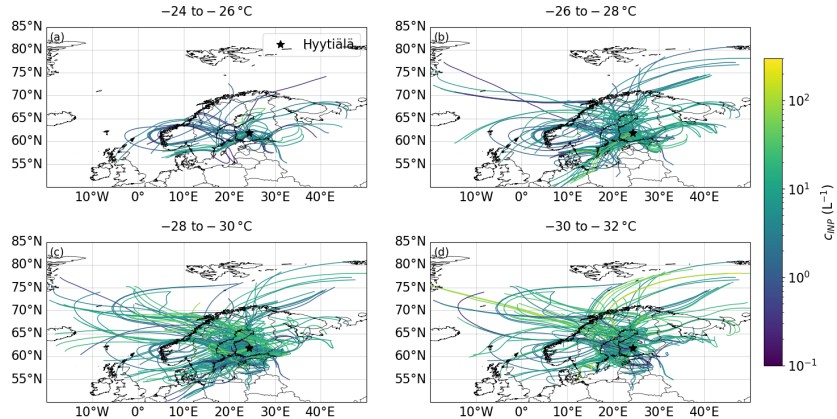

**Figure 11.** HYSPLIT 72 h backward trajectories calculated every 3 h. The colors represent the measured PINE INP concentration averaged over 3 h and then linked to the start time of the trajectory. The measurements are divided into four temperature ranges (a) to (d) covering 2 °C each.

the measured INP concentration may be explained by the enhanced release of biogenic aerosol. However, the relative humidity was nearly 100 % for most of the day.

The case studies presenting two selected time periods of 18 h and 22 h show a response of the INP concentration to different variables: case one more to meteorology and case two more to the ambient aerosol. This points out that on short time scales the INP concentrations can be driven by different factors, which leads to the correlation coefficients associated with the full time period being rather low.

## 3.6 Backward trajectory analysis

Backward trajectory analysis has been used in the past to identify scenarios that lead to enhanced INP concentrations or specific locations or processes that serve as sources of INP. For example, in a dataset of INP concentrations at the North Pole, Porter et al. (2022) used backward trajectory analysis to show that the periods of high INP concentrations corresponded to transport of air from the Russian coastline, whereas the lowest INP concentrations corresponded to air that had spent the preceding week circulating around the pack ice. For the PINE dataset presented here, we calculated backward trajectories using the HYSPLIT (Hybrid Single-Particle Lagrangian Integrated Trajectory) model (Stein et al., 2015), where one 72 h backward trajectory was calculated every 3 h. Each of the trajectories is color coded with the INP concentration that was measured at the start time of the trajectory. In order to extinguish a temperature dependency, the trajectories are divided into four temperature regimes, namely -24 to -26 °C, -26 to -28 °C, -28 to -30 °C and -30 to -32 °C (Figure 11). None of the four temperature bins shows a distinct pattern, i.e. air containing higher INP concentrations does not clearly differ from air masses containing

lower INP concentrations. The seemingly random distribution is consistent with the idea that sources of bioaerosol local to the measurement site control the INP population.

## 4    Conclusions

PINE is an instrument to measure ice-nucleating particles in an automated manner with a high time resolution and can be applied for both field and laboratory studies. In this paper, we used PINE to quantify the INP population in a finnish boreal forest during spring alongside an array of other INP and aerosol measurements. This is the first field deployment of PINE and we pay special attention to how PINE data can be analyzed and interpreted. This includes different time averaging methods, allowing us to quantify low INP concentrations and average over many expansions with confidence. The results show that the INP concentrations reported by PINE are consistent with other INP measurements. Overall these measurements show that the INP concentration in this location is up to three orders of magnitude greater than in remote ocean environments, but is generally lower or overlapping with other terrestrial environments, like the UK and the tropical Eastern Atlantic.

We use the high-time resolution INP data and aerosol measurements to challenge a number of parameterizations from the literature. We found that the Tobo 2013 (1) parameterization, based on measurements in a ponderosa pine forest in Colorado at 2370 m above sea level in summer, using the aerosol number concentration of particles larger than 0.5 μm in diameter as a predictive parameter is the most suitable predictor for our INP measurements. The fact that a parameterization based on measurements in a conifer forest in Colorado can predict the mean INP concentrations in a boreal forest in Finland implies some commonality between the two environments. Indeed, Tobo et al. (2013) concluded that biological particles contributed to the INP population in Colorado and we note that there is a moderate correlation between the fluorescent bioaerosol population and the INP concentration during HyICE-2018. The INP population in Colorado and Finland were also similar in that the DeMott 2010 parameterization produced a too shallow temperature dependence when compared to the respective datasets. However, a second parameterization proposed by Tobo et al. (2013), linking INP to fluorescent bioaerosol concentration, underpredicted our mean INP concentrations by about a factor of two. This indicates that the atmospheres of both environments have biological INPs that may have some commonality, but they are not identical.

Further evidence for the predominance of biological INP during HyICE-2018 across the full mixed-phase temperature regime comes from the study of Schneider et al. (2021) who found that INP at all temperatures (down to -25 °C in their study) were removed with heat treatments. This sensitivity to heat indicates the presence of protein based INPs consistent with those found in lichen, bacteria, fungus (Daily et al., 2022). The evidence presented here suggests that INPs in this boreal forest that are active below -24 °C can be also biological. In a previously published study from HyICE-2018 (Paramonov et al., 2020), using data from a thermal gradient diffusion chamber, it is suggested that INP were from distant sources. However, the positive correlation with time-over-land and the connection of the INP concentration to fluorescent particles detected in the F3 channel of WIBS, which can be assigned to NAD(P)H, a co-enzyme connected to the energy metabolism of cells, suggest that the boreal forest serves as a source of biogenic INP. It should also be borne in mind that Paramonov et al. (2020) made measurements from the end of February until beginning of April at a temperature of -30°C, hence only overlapping with our measurements

by 3 weeks. In the future we should aim to make INP measurements over a full annual cycle with PINE as the sources of INP will likley change with season.

Given that many surfaces were covered in snow during the first half of the HyICE-2018 campaign, including the tree canopy and the ground, several potential INP sources are not available. The snow melt did not bring any significant change in the INP concentration in the PINE dataset, suggesting that local INP sources are available independent of the snow. Potential reservoirs of INP that were exposed to air during HyICE-2018, despite the snow cover, are tree dwelling lichens (Proske et al., 2024) and the surface of Scots pine trees. It has been argued that lichens might produce atmospheric ice nucleating bioaerosol (Moffett et al., 2015), and it is known that the effect of humidity on bioaerosol production is complex with high humidity leading to production of structures on lichen that can become aerosolized later if the humidity decreases (Marshall, 1996; Armstrong, 1991; Proske et al., 2024). Also biological INPs on the surface of Scots pine trees, the dominant tree species at SMEAR II (Kokkila et al., 2002), were found to nucleate ice below a freezing temperature of -25 °C and could contribute to the INP population if there were an emission mechanism (Seifried et al., 2023). More work is clearly needed to understand the potential role lichen and other bioaerosol sources may play in the INP population in boreal forests, and perhaps wider afield.

*Data availability.* The INP data of PINE and NIPI are available at https://doi.org/10.5281/zenodo.10469663 (Brasseur et al., 2022). The INP data of INSEKT, the WIBS and APS and DMPS data are also presented in Schneider et al. (2021) and are available under https://doi.org/10.5445/IR/1000120666. The aerosol, trace gas and meteorological data are available at the SmartSMEAR data repository (https://avaa.tdata.fi/web/smart; Junninen et al. (2009).

*Author contributions.* FV, MPA and BJM wrote the paper. FV, MPA, LL, KH, JS, TS, BB and NSU conducted the measurements during the campaign. FV, JN, NB and RF supported the data treatment of the PINE measurements. MPA, PF, ADH, UP and GCEP contributed the results of the $\mu$l-NIPI data. PH and JK performed the WIBS measurements and analyzed the data. FV, MPA, LL, JS, ZB, EST, JD, TP, OM and BJM contributed to the discussion and interpretation of the results. OM, MK, TP and JD planned the HyICE-2018 campaign.

*Competing interests.* At least one of the (co-)authors is a member of the editorial board of Atmospheric Chemistry and Physics.

*Acknowledgements.* The authors gratefully acknowledge the technical staff of the Hyytiälä Forestry Field Station for their help and support during the HyICE-2018 campaign. Dr. Meri Räty is gratefully acknowledged for producing and sharing the time-over-land data set.

*Financial support.* This research has been supported by the Horizon 2020 (grant nos. ACTRIS-2 654109, ACTRIS PPP 739530, ACTRIS IMP 871115, 328616, ACTRIS-HY, ERA-PLANET 689443 and ACTRIS-CF 329274), the Helmholtz Association (grant no. 120101), the

KIT Technology Transfer (grant no. N059), the Academy of Finland (grant nos. 307331, 307537 and 286558), the European Research Council (grant no. MarineIce 648661), the Engineering and Physical Science Research Council (EP/S023593/1), the Natural Environment Research Council (NE/M010473/1, NE/L002574/1) , the University of Leeds International Strategy Fund , the Maj ja Tor Nesslingin Säätiö (grant no. 201900390), the Swedish Research Council (grant nos. 2013-05153 and 2017-00564) and the Alexander von Humboldt-Stiftung (grant no. 1188375). EST is supported by the Swedish Research Council VR(2020-03497).

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
