# Peer review of "Ice-nucleating particles active below -24 $^{\circ}\mathrm{C}$ in a Finnish boreal forest and their relationship to bioaerosols"

_EGUsphere, 2024_

## Referee Comment (RC1)

**Review:**

"Ice-nucleating particles active below -24 °C in a Finnish boreal forest and their relationship to bioaerosols"

**Summary:**

In their manuscript, Vogel and colleagues measured the ice nucleation activity of aerosols collected during a two-months field campaign in Hyytiälä, Finland, in spring 2018. They hypothesize that the measured ice-nucleating particles (INPs) active below -24°C originate from local biological sources. They measured INPs online, using a portable expansion chamber and compared it to offline droplet freezing measurements. The authors argue that the INPs below -24°C are derived from biological sources due to the alignment of their INP data with a parameterization of INPs in a pine forest in Colorado, a correlation with autofluorescing particles at sizes larger than 0.5 um and speculate that the negative correlation between relative humidity (RH) and INPs is connected to INPs originating from lichens. In addition, they observed no correlation between small particles and INP concentrations concluding that new particle formation is not the source of INPs.

Field studies like this are critical for the ice nucleation community. This study holds particular significance as biological INPs in large biosystems like the boreal forest are understudied, leaving much to be discovered. Thus, the work presented merits publication in Atmospheric Chemistry and Physics after addressing the following comments.

**GENERAL COMMENTS:**

- The manuscript lacks discussion on contrasting results from previous research, such as the study by Prenni et al., 2013, which indicates a correlation between increased biological INP concentration and high RH. The field study by Prenni et al., 2013 was conducted in a forest in Colorado, USA with the presence of pine trees as well. Since the authors argue in their manuscript that pine forests in Colorado and Finland can be compared (parameterization comparison), I miss the discussion about these contrasting results.

- Paramonov et al., 2020 measured INPs at the same research station in the same year and proposed that INPs were influenced by long-range transport, a result that is different from the outcome of this study. A more thorough discussion on the differences in outcomes between studies would enhance the manuscript.

- The authors speculate that the INPs originate from lichens. However, their argument, primarily based on the negative RH correlation and lichens not being covered by snow, warrants further elaboration. In general, biosystems are rather complex and the emission of bioparticles from the biosphere to the atmosphere depends on multiple factors (Pasanen et al., 1991). It is worthwhile noting that also certain fungal spores, like those from the genus *Cladosporium*, are released into the atmosphere at low RH (Sabariego et al., 2000). Further, *Cladosporium* have been documented to nucleate ice at temperatures consistent with those presented in this study (Iannone et al., 2011). Additionally, besides lichens being not covered by snow, INPs could also originate directly from trees. Seifried and Reyzek et al.,

2023 discovered biological INPs on the surface of Scots pines (bark and branches) capable of nucleating ice even below -25°C. Kokkila et al., 2002 mentions that Scots pine are the dominant tree species at the Hyytiälä research station. These points should be included in the discussion.

**SPECIFIC COMMENTS:**

*Abstract:*

- Line 6: To provide a clearer overview, I recommend inserting numbering (i), (ii), (iii) before each argument supporting the hypothesis that INPs below -24°C originate from biological sources.

*Introduction:*

- The Introduction is well written and the state of the art is well described.
- Line 37: Typo "relay"
- Line 50: Typo "bioaerosols" instead of bio aerosols

*Methods:*

- I suggest including a brief description or equation explaining how c_INP ($L^{-1}$) is calculated to ensure that readers understand it is measured per litre of air rather than per litre of liquid volume, as reported for some immersion freezing assays.
- Line 103: Typo "ETH" instead of EHT
- Line 122-124: missing space between units "L min$^{-1}$"
- Line 125: missing space between number and unit "6 min"
- Line 140: What is nanopure water? Please provide more details about the water used in the offline ice nucleation assays.
- Line 140: The freezing data is presented in °C in all graphs, so I recommend maintaining consistency by using the same temperature unit throughout the manuscript, including for the cooling rate. In addition, there is a space missing between the units "°C min$^{-1}$"
- Line 141: Consider rephrasing the sentence as not the well itself freezes but the content. Maybe: "[…] to a temperature at which all droplets freeze."
- Line 143: change unit for cooling rate
- Table 1: Last column, last cell: What do you mean by "end"? May 11, 2018?
- Line 156: add excitation and emission wavelengths

*Results & Discussion:*

- The months April and May are renowned for their elevated pollen concentrations, with species such as *Betula pendula* exhibiting notably high levels, especially towards the end of May in the southern regions of Finland (see e.g. Manninen et al., 2014). Could these pollen have contributed to the INP population?

- Figure 2, caption: Put listing items (a), (b), etc., before the sentence to match the style used in Figure 3.
- Figure 6 and line 284: It is not explained in the manuscript what FP3 for WIBS measurements is referred to. Please add the details.
- Line 285: missing space between the comma and "which"
- Line 295: consider rephrasing from "large concentrations" to "high concentrations"
- Line 316: Do you mean at around 14:00 (2 pm)? If so, use style hh:mm (same for time in line 335)
- Line 352: Typo "Figure 9" instead of "figure 9"

*Conclusion:*

- Are the aerosols after PINE collected and could be measured in the future with offline spectrum to obtain the entire freezing spectrum from the same aerosol population?

**References:**

Prenni, A. J., Tobo, Y., Garcia, E., DeMott, P. J., Huffman, J. A., McCluskey, C. S., ... & Pöschl, U. (2013). The impact of rain on ice nuclei populations at a forested site in Colorado. Geophysical Research Letters, 40(1), 227-231.

Paramonov, M., Drossaart van Dusseldorp, S., Gute, E., Abbatt, J. P., Heikkilä, P., Keskinen, J., ... & Kanji, Z. A. (2020). Condensation/immersion mode ice-nucleating particles in a boreal environment. Atmospheric Chemistry and Physics, 20(11), 6687-6706.

Pasanen, A. L., Pasanen, P., Jantunen, M. J., & Kalliokoski, P. (1991). Significance of air humidity and air velocity for fungal spore release into the air. Atmospheric Environment. Part A. General Topics, 25(2), 459-462.

Sabariego, S., Díaz de la Guardia, C. & Alba, F. (2000). The effect of meteorological factors on the daily variation of airborne fungal spores in Granada (southern Spain). Int J Biometeorol 44, 1–5. https://doi.org/10.1007/s004840050131

Iannone, R., Chernoff, D. I., Pringle, A., Martin, S. T., & Bertram, A. K. (2011). The ice nucleation ability of one of the most abundant types of fungal spores found in the atmosphere. Atmospheric Chemistry and Physics, 11(3), 1191-1201.

Seifried, T. M., Reyzek, F., Bieber, P., & Grothe, H. (2023). Scots Pines (Pinus sylvestris) as Sources of Biological Ice-Nucleating Macromolecules (INMs). Atmosphere, 14(2), 266.

Kokkila, T., Makela, A., & Nikinmaa, E. (2002). A method for generating stand structures using Gibbs marked point process. Silva Fennica, 36(1), 265-277.

Manninen, H. E., Bäck, J., Sihto-Nissilä, S. L., Huffman, J. A., Pessi, A. M., Hiltunen, V., ... & Petäjä, T. (2014). Patterns in airborne pollen and other primary biological aerosol particles (PBAP), and their contribution to aerosol mass and number in a boreal forest. Boreal Environment Research, 19, 383.

---

## Author Comment (AC1)

**Author's response to Referee #1**

We thank reviewer #1 for the thoughtful comments on our manuscript. Below we provide detailed answers to the individual point giving in black the original comments, in red our answer and in blue the changes in the manuscript.

**General comments**

1) The manuscript lacks discussion on contrasting results from previous research, such as the study by Prenni et al., 2013, which indicates a correlation between increased biological INP concentration and high RH. The field study by Prenni et al., 2013 was conducted in a forest in Colorado, USA with the presence of pine trees as well. Since the authors argue in their manuscript that pine forests in Colorado and Finland can be compared (parameterization comparison), I miss the discussion about these contrasting results.

We agree with the reviewer, that a discussion on these contrasting results is missing (although we did discuss related studies to that of Prenni). Also reviewer #2 commented on that, and we included a discussion on that in the manuscript. This is part of the discussion mentioned under point (3) and the updated manuscript can be found there.

2) Paramonov et al., 2020 measured INPs at the same research station in the same year and proposed that INPs were influenced by long-range transport, a result that is different from the outcome of this study. A more thorough discussion on the differences in outcomes between studies would enhance the manuscript.

The study of Paramonov et al., 2020 presents INP measurement from the end of February until beginning of April at a temperature of -30°C. With that the overlapping period was 3 weeks. Both studies consider the entirety of their measurements, meaning that for Paramonov et al., 2020 there is a significant contribution to the total INP spectrum before our measurements started and our manuscript contains more than one month additional data for a time period not covered in the other study. Paramonov et al., 2020 says that 'No single dominant local or regional sources of INPs in the boreal environment of southern Finland could be identified', from which they conclude that the INP population at SMEAR II originates from long-range transport. In our study, we also did not identify one single dominant local source, however we provide evidence for periods of a more local influence of INPs, such as a correlation between INP and time over land of 0.39 and a connection of the INP concentration to fluorescent particles. Therefore, we draw a different conclusion for our study. We already pointed to this difference in the conclusions, but extended the statement as follows:

L.381 ff: In a previously published study from HyICE-2018 (Paramonov et al. 2020), using data from a thermal gradient diffusion chamber, it is suggested that INP were from distant sources. However, the positive correlation with time-over-land and the connection of the INP concentration to fluorescent particles detected in the F3 channel of WIBS, which can be assigned to NAD(P)H, a co-enzyme connected to the energy metabolism of cells, suggest that the boreal forest serves as a source of biogenic INP. It should also be borne in mind that (Paramonov et al., 2020) made measurements from the end of February until beginning of April at a temperature of -30°C, hence

only overlapping with our measurements by 3 weeks. In the future we should aim to make INP measurements over a full annual cycle with PINE as the sources of INP will likely change with season.

3) The authors speculate that the INPs originate from lichens. However, their argument, primarily based on the negative RH correlation and lichens not being covered by snow, warrants further elaboration. In general, biosystems are rather complex and the emission of bioparticles from the biosphere to the atmosphere depends on multiple factors (Pasanen et al., 1991). It is worthwhile noting that also certain fungal spores, like those from the genus Cladosporium, are released into the atmosphere at low RH (Sabariego et al., 2000). Further, Cladosporium have been documented to nucleate ice at temperatures consistent with those presented in this study (Iannone et al., 2011). Additionally, besides lichens being not covered by snow, INPs could also originate directly from trees. Seifried and Reyzek et al., 2023 discovered biological INPs on the surface of Scots pines (bark and branches) capable of nucleating ice even below -25°C. Kokkila et al., 2002 mentions that Scots pine are the dominant tree species at the Hyytiälä research station. These points should be included in the discussion.

Indeed, the emission of bioaerosol is quite complex and is based on multiple factors. As also mentioned be referee #2, we extended our discussion around the negative correlation of INP and RH and included a new subsection. Also, the referee suggested the example of Cladosporium from Sabariego et al. We decided not to include this particular example in part because it is not clear how relevant Cladosporium is for Boreal forests (Saberiego et al. is a study in Spain), but also while there was a correlation reported with temperature, the correlation with RH was actually rather weak (and mixed sign, depending on type). We added the following new section:

3.4.2 The link between high INP concentrations and low relative humidity

[revised manuscript text omitted]

We further followed the suggestion of including the discussion of more bioaerosol sources and adjusted our manuscript as follows:
L. 306: Another study suggests the surface of Scots pine trees, the predominant tree species in Hyytiälä (Kokkila et al., 2002) as a potential source of INPs (Seifried et al., 2023). Since lichens and the bark and branches of trees are one of the few biological entities L. 386: Potential reservoirs of INP that were exposed to air during HyICE-2018, despite the snow cover, are tree dwelling lichens (Proske et al., 2024) and the surface of Scots pine trees. L. 390 Also biological INPs emitted from the

surface of Scots pine trees, the dominant tree species at SMEAR II (Kokkila et al., 2002), were found to nucleate ice below a freezing temperature of -25 °C and could contribute to the INP population (Seifried et al., 2023).

**Specific comments**

4) Line 6: To provide a clearer overview, I recommend inserting numbering (i), (ii), (iii) before each argument supporting the hypothesis that INPs below -24°C originate from biological sources.

We added the numbering before the three key arguments.

...this location are also from biological sources: (i) an INP parameterization developed for a pine forest site in Colorado, where many INPs were shown to be biological, produced a good fit to our measurements; a moderate correlation of INP with aerosol concentration larger than 0.5 µm and the fluorescent bioaerosol concentration; (ii) a negative correlation with relative humidity that may relate to enhanced release of bioaerosol at low humidity from local sources such as the prolific lichen population in boreal forests. (iii) The absence of correlation...

5) The Introduction is well written and the state of the art is well described.

We are pleased the referee found the intro well-written

6) Line 37: Typo "relay"

Changed to 'rely'

7) Line 50: Typo "bioaerosols" instead of bio aerosols

Changed to 'bioaerosols'

8) I suggest including a brief description or equation explaining how c_INP ($L^{-1}$) is calculated to ensure that readers understand it is measured per litre of air rather than per litre of liquid volume, as reported for some immersion freezing assays.

We inserted an equation showing how the INP concentration is calculated and a respective description in L. 142ff.

The resulting INP concentration is calculated per liter of air following equation 1:

$c_{n,INP} = V_{sol} / V_{air} * c_{n,INP,sol} = - V_{sol} / V_{air} * d / V_{well} * \ln(f_{liq}(T))$ . (1)

d is the dilution factor, which is for this analysis 1, 10 or 100, $V_{air}$ is the volume of the sampled air, that passed the filter and $f_{liq}$ is the fraction of liquid wells at a certain temperature of the measurement.

In L. 144, we added an additional sentence for the µL-NIPI.

The obtained INP concentrations are given per standard liter of air

9) Line 103: Typo "ETH" instead of EHT

Changes to 'ETH'

10) Line 122-124: missing space between units "L min -1 "

Space added between 'L' and 'min-1' in all three lines

11) Line 125: missing space between number and unit "6 min"
Space added

12) Line 140: What is nanopure water? Please provide more details about the water used in the offline ice nucleation assays
NanopureTM water is filtered and deionized water. Typically, the purity of water is given with its conductivity, so we added this value in the text in L. 136.
... 8 ml of nanopureTM (conductivity of approximately 0.056 - 0.057 µS cm−1) water ...

13) Line 140: The freezing data is presented in °C in all graphs, so I recommend maintaining consistency by using the same temperature unit throughout the manuscript, including for the cooling rate. In addition, there is a space missing between the units "°C min-1 "
We changed the units of the cooling rate in L. 140 and L. 143 from K min-1 to °C min-1

14) Line 141: Consider rephrasing the sentence as not the well itself freezes but the content. Maybe: "[…] to a temperature at which all droplets freeze."
We agree that this sentence sounds misleading. We rephrased it in the text.
... to a temperature at which all solution droplets inside the wells freeze.

15) Line 143: change unit for cooling rate
We changed the cooling rate as suggested

16) Table 1: Last column, last cell: What do you mean by "end"? May 11, 2018?
We intended to say that it was measuring until the end of the campaign, which was after May 11, 2018. However, we agree to change to the end date of PINE measurements to avoid misunderstandings.

(17) Line 156: add excitation and emission wavelengths
We added the two excitation and detection band wavelengths in L. 156
... measures particle fluorescence with two excitation lasers (wavelength of 280 nm and 370 nm), and emission is monitored in two detection bands (310 nm to 400 nm and 420 nm to 650 nm).

18) The months April and May are renowned for their elevated pollen concentrations, with species such as Betula pendula exhibiting notably high levels, especially towards the end of May in the southern regions of Finland (see e.g. Manninen et al., 2014). Could these pollen have contributed to the INP population?
The winter in Hyytiala in 2018 lasted rather long and trees started having leaves only at the beginning of May. The concentration of fluorescent particles increased already towards the end of March as shown in Schneider et al. 2021. However, the INP concentration did not show any significant increase. It is possible that the pollen contributed to the INP population, but we do not

have any further poof for that. Since pollen is not sensitive to the wet heat test, the heat sensitivity in droplet freezing assays is indicative of something other than pollen (but does not rule out pollen).

19) Figure 2, caption: Put listing items (a), (b), etc., before the sentence to match the style used in Figure 3.
We moved the listing items in the figure caption from the end of the sentence to the front of the sentence

20) Figure 6 and line 284: It is not explained in the manuscript what FP3 for WIBS measurements is referred to. Please add the details.
We added an explanation of the FP3 channel and gave indications which type of biological particles can be detected in this specific channel.
The p values indicate that these correlations are significant. An aerosol particle entering WIBS gets excited by two lasers with a different wavelength and can be detected in two different detection bands. Depending on its composition, fluorescence is detected in only one or more laser detection band channels. The F3 channel is fluorescence triggered by the 370 nm laser and detected in the 420 nm to 650 nm band. Particles detected only in this channel can be assigned to NAD(P)H, which is a tracer for the viable biological fraction (Savage et al., 2017).

21) Line 285: missing space between the comma and "which"
We rephrased L.285 ff based on the comment before. Therefore, this part with the missing comma was deleted

22) Line 295: consider rephrasing from "large concentrations" to "high concentrations"
Changed as suggested

23) Line 316: Do you mean at around 14:00 (2 pm)? If so, use style hh:mm (same for time in line 335)
Changed as suggested

24) Line 352: Typo "Figure 9" instead of "figure 9"
Changed as suggested

25) Are the aerosols after PINE collected and could be measured in the future with offline spectrum to obtain the entire freezing spectrum from the same aerosol population?

Aerosol particles exiting PINE are currently not collected on filters or as a bulk. This is something that would be interesting to work on in the future, perhaps even with a virtual impactor.

---

## Author Comment (AC2)

**Author's response to Referee #2**

We thank reviewer #2 for the valuable comments on our manuscript. Below we provide detailed answers to the individual point giving in black the original comments, in red our answer and in blue the changes in the manuscript.

**Specific comments**

1) The time series of INP data measured using the PINE chamber are unique and potentially valuable. However, this work appears to repeat essentially the same work as reported by Schneider et al. (2021) and Brasseur et al. (2022). I strongly encourage that the authors reconsider if they can add new topics and findings using their data sets. Then, the authors should significantly update the contents of the manuscript in order to clarify the difference between this study and other earlier studies.

We disagree that we report that same work as the two previous papers. Schneider et al., 2021 measured at temperatures higher than -25°C, where it is known from literature that the INP population is often dominated by biogenic aerosol in terrestrial environments. Furthermore, they present daily filter measurements to evaluate a day-by-day and also seasonal variability. In our manuscript we present measurement at temperatures lower than -24°C, the temperature range that was not covered by Schneider et al., 2021 and our data are given in a time resolution of 6 min, allowing us to average over 1h and by that investigate an hour-by-hour variability, which is driven by different factors than the long-term variability. The different driving factors are highlighted by (1) the fact that the Schneider et al., 2021 parameterization does not fully represent our data, and overestimates the INP concentration especially for lower temperatures that were not covered by their study and (2) very low correlation coefficient with the ambient air temperature, which Schneider et al., 2021 showed to be high for their measurements. Brasseur et al., 2022 compare four selected days, for a time of maximum 12h, the INP concentrations measured by the various instruments with various parameterizations. Their Measurement Report was meant to give an overview of the campaign rather than a full analysis. We apply the parameterization on two months of continuous measurements and by that present a wider variability. For this reason, we think that it is essential to provide the evaluation of the parameterizations on the full dataset. In order to emphasize the importance of our manuscript, we added a paragraph at the end of the introduction in L. 60:

The study on the same campaign presented by Schneider et al. (2021) focused on the temperature range higher than -25 °C and seasonality, while here we present measurements at temperatures below -24 °C and investigate the hourly variability. Paramonov et al. (2020) performed measurements in the same temperature range, but over a different time period. Their measurements were from end of February 2018 to beginning of April 2018, whereas ours were from mid-March until mid May, including the time of snow melt and early spring. Only

four days of our measurements are included in the measurement report from Brasseur et al. (2022), challenging several literature parameterizations. In our present manuscript, we extend this analysis for the full 2 months to better understand the variability of the dataset.

2) In my understanding, cloud-chamber-type instruments other than PINE (e.g., SPIN, PINC) had also been used to measure the time series of INPs active below -24 degree C at this site during the HyICE-2018 campaign. The authors should compare the difference between the PINE and the other instruments in the lower temperature regime and evaluate the strengths (and limitations) of the PINE. For example, to my knowledge, Paramonov et al. (2020) already reported the full INP data sets active around -30 degree C measured using a PINC during the overlapping period.

During the HyICE-2018, PINE, PINC and SPIN measured the INP concentration at temperatures lower than -24°C. However, there are substantial differences in the data coverage and availability so there we have not done a full comparison on all data. There were a few occasions within HyICE-2018 where we purposefully performed intercomparison tests where we all aimed at reporting in overlapping temperatures and times. These are reported in Brasseur et al. (2022) and it would not be appropriate to repeat those intercomparisons here. That said, we have added the data from PINC in figure 2a based on comment 13. SPIN covered the same time period as PINE, but data are only available for the 2 days that were part of the intercomparison of Brasseur et al., 2022. Other data are not publicly available. PINC measured from end of February until April 2 and by that covered the time period where everything was covered in snow. In contrast to that, PINE provided data during the snow covered period, the transition period and the snow free period. By that PINE covers more variability in potential aerosol sources, especially from local emissions. As mentioned in our answer to comment 1), we provide a detailed discussion in the revised manuscript. We also added the data of PINC in figure 2 and changed the according discussion. We don't think that it is valuable to add the SPIN data in this figure, because the two available days are not representative.

L. 168 ... and CFDC measurements of PINC from the same campaign (Figure 2a)...

L. 172: PINE and PINC data overlap at the lower end of the covered temperature range and fully agree in the span of measured INP concentration.

3) Please explain more details for the WIBS data. It is unclear how the WIBS fluorescent particle data were applied to the Tobo et al. (2013) (2) parameterization. As a result, it is difficult to judge whether the authors' explanations that "this means that the measured INP concentration ~ from different sources (Lines 250-252)" and "a second parameterization proposed by Tobo et al. (2013), linking ~ but they are not identical (Lines 373-376)" are indeed reasonable. For example, Tobo et al. (2013) used the data on fluorescent particles larger than 0.5 um measured using UV-APS that employs an excitation wavelength of 355 nm and detects laser-induced

fluorescence spectra in the range of 420–575 nm). Did they use the same setup as the UV-APS?
The WIBS-neo used for the measurements of fluorescent particles during HyICE-2018, has two excitation laser and two detection bands. The wavelength of laser 2 (370nm) is close to the wavelength of the UV-APS laser. The detection band of the UV-APS falls within the second detection band of the WIBS of 420nm – 650nm, but is missing the higher wavelengths. The size cut of both instruments is 0.5 um. We are aware, that the two instruments have their differences, but we still think that it is worth to challenge the Tobo 2013 parameterization based on the total concentration of fluorescent particles, since the measurement environments have similarities. We added in the methods section more details of the WIBS, and discuss this specific parameterization keeping the differences between the two instruments in mind.
L. 150 ff: WIBS (Wideband integrated bioaerosol sensor) measures particle fluorescence with two excitation lasers (wavelength of 280 nm and 370 nm), and emission is monitored in two detection bands (310 nm to 400 nm and 420 nm to 650 nm). ...
L. 158: WIBS detected particles with an optical diameter larger than 0.5 μm.
L. 226: ... Tobo 2013 (2) uses the concentration of fluorescent particles with a diameter larger than 0.5 μm measured with a UV-APS with an excitation wavelength of 355 nm and an emission wavelength range between 420 nm and 575 nm ...
L. 228: Even though fluorescent particle measurements to develop the Tobo 2013 (2) parameterization were done with a different instrument, it is interesting to apply this parameterization on our data due to the similarity of the measurement environments. Due to the setup of WIBS it provides measurement of seven different combinations of excitation wavelength and detection band wavelength, whereas the UV-APS provides only one channel. By that the WIBS can capture more various bioaerosols and the INP concentrations predicted by Tobo 2013 (2) should be seen as an upper estimation.
L. 249: As discussed before, this is the upper estimated of the INP concentration due to the differences in UV-APS and WIBS, and the predicted INP concentrations can be even lower.

4) Brasseur et al. (2022) announced that "forthcoming studies would explore atmospheric vertical profiles of INPs, INP sources and transport modeling, plausible links between INP abundance and NPF, and the ice nucleation activity of boreal biology such as flora and fungi". If this work is a part of the forthcoming studies, the authors should present some more details for the comparison of INPs with NPF. Since the information on the correlation coefficients only (Figure 6) is obviously insufficient, the authors should show the time series of the INP and NPF data.

We originally took the approach of keeping the manuscript succinct, but see the value of this additional analysis.  We have expanded on this topic with an additional section and an additional figure:

**3.4.1 INP concentrations during new particle formation events**

[Figure]

**Figure 7.** Overall comparison of the INP concentration during NPF events and outside NPF events together with three case studies. (a) Boxplot of the INP concentration on days flagged NPF event days and days without an NPF event. The borders of the box represent the 25th and 75th percentile, the horizontal line inside the box the median, the whiskers the 5th and 95th percentile and the single points all the data outside of that. (b), (c) and (d) are three cases (March 16, April 10 and May 5) of NPF events of category Ib, where the background of each panel shows the particle number size distribution, with the diameter in the left y-axis and dN/dlogDp as the colorbar. The first right y-axis is the INP concentration and the second right y-axis the aerosol number concentration measured by the DMPS and APS. Please note that the y-axis of the INP concentration varies, due to the high variability of INP.

The low correlation between INPs and the concentration of small particles (Fig.6) suggests that particles formed during new particle formation (NPF) events do not contribute to the INP population. During NPF events, gaseous species nucleate to form a critical cluster which then grows via gas to particle conversion to particles of tens of nanometers in diameter over the course of hours Kulmala et al., 2001. These newly formed particles are very high in number concentration and can be detected based and the aerosol size distribution and the increase in aerosol number concentration. SMEAR II is known for its NPF events and the analysis and categorization of these events is based on Dal Maso et al., 2005, who introduced category Ia, Ib

and II events, which designates the intensity of an event. The occurrence of NPF days during HyICE-2018 was already presented in Brasseur et al., 2022, where individual days were flagged as NPF days, independent of their category. To investigate the connection between NPF events and INPs, we flagged the hourly INP data measured with PINE at a freezing temperature between -27 °C and -30 °C with the different NPF event categories. In case none of the three NPF categories was detected, we set the flag to no NPF event. By comparing the INP concentration on NPF event days (independent of the category) and non-NPF event days (Fig.7a), a similar median and variability of the INP concentration is seen in the two boxes. This indicates that NPF events have no effect on the INP population.

In order to explore the NPF events in more detail, we plot time series of PINE INP measurements alongside aerosol concentrations and the particle number size distribution on three days, one from each month of the campaign. The case on March 16 (Fig.7b) shows a steep increase in the aerosol number concentration at 12:00 local time from about 2000 cm-3 to approximately 8500 cm-3, which characterizes the start of a NPF event, which is also visible in the high concentration of small particles at the same time. The aerosol concentration decreases to the initial value at 23:00 local time. During the whole day the INP concentration is scattered between 2 L-1 and 12 L-1 without showing a distinct pattern. In the other two cases on April 10 and May 5 (Fig.7c and d) the aerosol concentration also shows a steep increase when NPF sets in and lower aerosol concentration outside the NPF event. In these two cases the INP concentration reaches higher values during phases of lower aerosol concentrations and drops when NPF is initiated. This may be because NPF events are favored when the aerosol surface area concentration is low, which is also when the INP concentrations would generally be expected to be lower. This analysis strengthen the conclusion that particles formed during NPF events in the size range up to 50 nm in diameter do not contribute to the INP population.

5) The possible links between INPs and low RH are potentially interesting. On the other hand, many earlier studies have reported elevated INP concentrations under high RH conditions at a forested site in Colorado (Huffmann et al., 2013; Prenni et al., 2013; Tobo et al., 2013) and other locations (Wright et al., 2014). I would suggest showing figure(s) that could explain the relationship between INP and RH data. Then, please discuss the discrepancy between this study and the above studies if there was indeed a clear negative correlation between the INP and RH data.

We have added an addition section to cover this in more detail and place the result in the context of the literature:

3.4.2 The link between high INP concentrations and low relative humidity

[revised manuscript text omitted]

**Technical comments**

6) I would suggest changing the title appropriately, because other related literatures (Paramonov et al. 2020; Brasseur et al., 2023) have already reported INP data below -24 degree C during the HyICE-2014.
Our strong preference is to keep the title. It contrasts strongly with the title used by Paramonov (Condensation/immersion mode ice-nucleating particles in a boreal environment) and emphasizes the different conclusion (i.e. bioaerosols). It is also very different to the title used by Brasseur: 'Measurement report: Introduction to the HyICE-2018 campaign for measurements of ice-nucleating particles and instrument inter-comparison in the Hyytiälä boreal forest'

7) Please indicate the full name and the abbreviated name in the same manner (e.g., see "Portable Ice Nucealtion Experiment (PINE)" in Line 59 and "PINC (Portable Ice Nucleation Chamber)" in Line 104.
We changed the abbreviations of PINC and PINCii in L.103-104 as suggested
... Portable Ice Nucleation Chamber (PINC, operated by ETH Zurich; Kanji et al. (2013)) and Portable Ice Nucleation Chamber ii (PINCii, operated by University of Helsinki and University of Gothenburg ...

8) Please explain some more details for the experimental setup for the PINE. For example, what is the cut-off diameter of the inlet?

Section 2.2 contains the results of the inlet characterization of PINE in L. 107 ff, that were performed during the campaign. Also, the overview paper (Brassuer et al., 2022) contains many of the details around the experimental setup. We rephrased the pertinent sentence to that points out the cut-off diameter test was done specifically for the PINE inlet and not for the inlet in general.

L. 108: An additional impactor was not installed. The PINE inlet was characterized onsite using an OPC (MetOne, GT 526S) directly at the aerosol inlet and at the entrance of PINE to compare the number concentration in five different size bins of particles behind the inlet with those entering PINE. With that the transmission efficiency was calculated, which showed ...

9) I assume that the PINE uses an OPC that can detect larger aerosol particles, liquid cloud droplets and ice crystals as single particles (Line 84), but the size range covered by this OPC is not clear. It is also unclear how the PINE used here discriminated ice crystal signals from ambient aerosol particles and liquid droplets?

We agree with the reviewer, that additional information is valuable for the reader and added them in L. 85 ff.

The OPC detects larger aerosol particles, liquid cloud droplets and ice crystals as single particles, based on the size and shape dependent scattering signal. Due to their spherical shape, liquid cloud droplets are detected with their actual size, while the aspheric ice crystals show a higher scattering signal and are thus detected a larger diameter than cloud droplets. Aerosol particles have been proved to be detected at smaller diameters than ice crystals and can thus be differentiated from ice crystals. A detailed discussion can be found in Möhler et al. (2021). During the HyICE-2018, the OPC was set to measure particles with a diameter between 0.6 μm and 40 μm.

10) Lines 96-97: Please explain if the INP number concentrations presented here were corrected to the values at standard temperature and humidity conditions (0 degree C and 1 atom) or not.

The INP concentrations presented in the manuscript are given in $stdL^{-1}$. We added a sentence in L. 97.

... and corrected for standard conditions (i.e. all ice-particle concentrations are reported at standard temperature and pressure).

11) Line 99: What do you mean by "the uncertainty for PINE is given as 20 %"?

We intended to say that the uncertainty of the INP concentration is given as 20%, with further explanation given in Möhler et al., 2021. The sentence in L. 99 was adjusted to the following:

The uncertainty for the INP concentration is given as 20 % (Möhler et al., 2021) and is not displayed in the following figures to keep them clear.

12) Lines 111-115: Please specify the RH setup for mixed-phase cloud conditions in the PINE chamber.

We have added the following to explain that the RH in the chamber is defined by the presence of cloud droplets.

L. 115 ff: In this campaign we operated PINE under mixed-phase conditions, that is we always generated a liquid cloud and counted the number of ice crystals that grew out of the liquid cloud. During an expansion, the temperature inside the cloud chamber is lowered around 6 °C within 40 s. It is evident that water saturated conditions are reached with the formation of a liquid cloud.  Any ice crystals that form during the expansion grow rapidly in the strongly ice-supersaturated environment.  In contrast to other instruments, the relative humidity during the expansion was not directly measured since the appearance of droplets defines the RH in the chamber.

13) Figure 2a: I would suggest including INP data derived from SPIN and PINC during the HyICE-2014 campaign.

The PINC data are available in Paramonov et al., 2020, so we included them as a grey bar in figure 2a. SPIN data are only available for a few hours for two days from Brasseur et al., 2022, so they do not represent the full range of INP concentrations measured with SPIN.

14) Lines 262-264: Please explain how the PINE chamber captures deposition and condensation nucleation in the Methods section.

Deposition nucleation occurs while water saturation is not exceeded. In the case of the measurements presented here, this only includes a few seconds before cloud formation sets in and would be seen as an ice signal before the formation of the liquid cloud. While in principle PINE can be used below water saturation, we do not do it here hence have not included an explanation in the methods. In the statement in L. 262-264, we wanted to imply that with PINE we are not limited to one freezing mechanism in contrast to INSEKT.

15) Lines 271-272: Please revise this description if the temperatures were between -27 and -30 degree C as explained in the Figure 6 caption.

We changed the sentence in L. 271-272 to the following:

To further investigate potential connections between ice nucleation activity and meteorological variables and aerosol properties at temperatures between -27 °C and -30 °C, correlation coefficients are calculated.

16) Line 284: What do you mean by "FP3 channel"?  Please also check my comment 3.

Also reviewer #1 commented on that and we inserted an explanation in L. 287 ff.

An aerosol particle entering WIBS gets excited by two lasers with a different wavelength and can be detected in two different detection bands. Depending on its composition, fluorescence is detected in only one or more laser detection band channels. The F3 channel is fluorescence

triggered by the 370 nm laser and detected in the 420 nm to 650 nm band. Particles detected only in this channel can be assigned to NAD(P)H, which is a tracer for the viable biological fraction (Savage et al., 2017).